# Impact of digital finance on enterprise green innovation: From the perspective of information asymmetry, consumer demand and factor market distortions

**Linzhi Han, Zhongan Zhang** [ORCID] *

School of Economics and Management, Xinjiang University, Urumqi, China

* 20200500214@stu.xju.edu.cn

## Abstract

The study endeavors to empirically assess the influence of digital finance on promoting enterprise green innovation, while simultaneously probing its underlying mechanisms, by leveraging panel data from a sample of 2071 China A-share listed firms over an extensive time frame spanning from 2011 to 2021. The findings demonstrate that digital finance plays a crucial role in promoting enterprise green innovation, and that both the coverage breadth and usage depth of digital finance have a significant effect on enterprise green innovation, but the digitization level of digital finance also has a non-significant effect on enterprise green innovation, and the conclusions hold even after multiple robustness tests and consideration of endogeneity issues. Furthermore, heterogeneity analysis reveals that digital finance is only has a significant promoting effect on green innovation of SMEs, and high-tech enterprises. After conducting the mechanism analysis, it has been noted that digital finance serves as a facilitator in promoting green innovation in enterprises by reducing information asymmetry, stimulating consumer demand, and attenuating the distortion of regional factor markets. Intellectual property protection and environmental governance will strengthen the positive impact of digital finance on enterprises' green innovation. The research results provide policy implications for the green development of digital finance enabling enterprises.

## 1. Introduction

Since implementing economic reforms and opening up its markets, China has experienced significant growth in its economy, but its extensive economic development model has brought severe challenges to the ecosystem and limited the economy's development towards high-quality standards [1–3]. In light of the escalating and pressing environmental concerns that continue to persist, it was during the 75th session of the United Nations General Assembly in September 2020 that delegates from the Chinese government proffered their proposal to accomplish the pinnacle of carbon dioxide emissions by the year 2030, whilst simultaneously striving to attain carbon neutrality by the year 2060 [4–6], further increasing the research

---

**Data Availability Statement:** Publicly available datasets were analyzed in this study. The data can be found at: The Peking University Digital Financial Inclusion Index of China (idf.pku.edu.cn), The

CNRDS Database (www.cnrds.com), China Stock Market & Accounting Research Database (www.csmar.com).

**Funding:** This research was funded by Program of National Social Science Foundation of China (grant number 22BJL061), Major Project of Xinjiang Social Science Foundation (grant number 21AZD008) and the 2022 National Undergraduate Innovation Training Program of Xinjiang University "Re-search on the Impact of Digital Inclusive Finance on Enterprise Green Technology Innovation" (grant number 202210755045).The funders had no role in study design, data collection and analysis, decision to publish, or preparation of the manuscript.

**Competing interests:** The authors have declared that no competing interests exist.

enthusiasm related to green innovation. Enterprises are the users of natural resources and the implementers of green innovation, and it is an inevitable choice for the sustainable development of all enterprises to give full play to the main position of green innovation, enhance the innovation ability of green and low-carbon science and technology, and accelerate the research and development and application of advanced technologies for energy conservation and carbon reduction [7–10]. However, enterprise green innovation projects require more investment funds, have high risk and a long return period, and low cost and efficient capital guarantees are necessary conditions [11, 12]. However, traditional finance has many problems, such as cumbersome procedures, high thresholds, and low efficiency for financing for green innovation projects, small and medium-sized enterprises, in particular, can impede the drive towards green innovation among businesses [13–15]. Over the past few years, the emergence of digital finance rooted in big data, cloud computing, and artificial intelligence has experienced a significant boom [16, 17], Due to its characteristics of wide coverage, deep application, and digitalization, it has become an important means of solving the problems of enterprise financing difficulties and correcting the mismatch of financial information. How digital finance affects the green innovation of enterprises has aroused great attention from academia and policy levels, as it can provide empirical evidence for digital finance to better support the green innovation of the real economy by exploring its effects and mechanisms.

Using the aforementioned context, this article methodically clarifies the theoretical mechanism through which digital finance impacts the green innovation of enterprises. A dataset comprising 2071 A-share listed companies, China was utilized from the period spanning 2011 to 2021, and the study analyzed the effects and mechanisms of digital finance, including its sub-dimensional indicators, on the promotion of green innovation among enterprises.

The main contributions of this study to the existing literature are summarized as follows: First, most existing related literatures have explored the mechanism of digital finance on enterprise green innovation from the perspectives of alleviating financing constraint, enhancing financial flexibility, solving internal and external information constraint, improving environmental information disclosure quality, improving corporate transparency, mitigating financial mismatch, increasing R&D investment, and improving internal control, whereas our study theoretically analyzes and empirically examines the mechanisms of information asymmetry reduction, consumer demand stimulus and factor market distortion mitigation by which digital finance affects enterprise green innovation, as well as the impacts of intellectual property protection and environmental governance on the relationship between digital finance and enterprise green innovation, which sheds new light on digital finance influencing enterprise green innovation. Second, most existing studies on the heterogeneity of the impact of digital finance on enterprise green innovation are based on the level of regional economic development, the degree of pollution in the industry, the nature of enterprise property rights, the life cycle of the enterprise, the level of regional financial development and the intensity of regional financial regulation, whereas our study analyzes the heterogeneity of the results of the study from the perspectives of enterprise scale and whether they possess high-tech qualifications, and enriches the study on the impact of digital finance on the green innovation of enterprises with different characteristics.

The paper is structured as follows: Section 2: Literature review presents a review of relevant literature and highlights marginal contributions. Section 3: Theoretical analysis and research hypothesis detail our theoretical analysis and hypothesis. Section 4: Analysis of the results of the return discusses how digital finance impacts enterprise green innovation. Section 5: Heterogeneity analysis and mechanism test tests the impact mechanism and heterogeneity of digital finance on enterprise green innovation. Section 6: Conclusions and policy implications, the concluding section offers policy recommendations based on the findings of this study.

## 2. Literature review

In recent years, more and more scholars pay attention to the impact of digital finance on green innovation, some literatures have investigated the mechanism of digital finance on corporate green innovation, which mainly includes the mechanism of financing constraint alleviation, R&D investment increase, financial flexibility en-hancement, environmental information disclosure quality improvement, financial mismatch mitigation, internal control improvement and internal and external information constraint dissolution, etc. Liu et al. (2022) [18] found that the development of digital finance, its coverage breadth and usage depth will increase the number of green patents granted by enterprises, especially the number of green invention patents granted, and significantly increase the quantity and quality of green innovations, and this effect is more obvious in the economically backward areas and highly polluted industries, digital finance promotes green innovation by alleviating capital constraints and increasing R&D investment. Fan et al. (2022) [19] found that digital finance and its coverage breadth and usage depth effectively promote corporate green innovation, but the degree of digital finance digitization has no significant effect on corporate green technology innovation, and digital finance only has a significant positive impact on the green technology innovation of enterprises with high financing constraints and high financial leverage groups, industries with low concentration, growth, and low quality of environmental disclosure reports and environmental governance reports; digital finance can improve internal financing for corporate green technology innovation by reducing financing costs and increasing financial flexibility. Liu et al. (2022) [20] found that digital finance can stimulate enterprises' green innovation by increasing the coverage breadth and usage depth of digital finance, and the impact is stronger when the analyst optimism bias is lower and the synchronization is higher; digital finance can reduce the financial constraints of enterprises by providing them with loans and improving cash flow, improve transparency by addressing external information constraints, and increase internal control and re-search investment, thus making enterprises more and more willing to carry out green innovation. Kong et al. (2022) [21] found that digital financial institutions can alleviate the information asymmetry in the green innovation market and directly promote the green innovation behavior of enterprises through digital technologies such as big data analysis of enterprise behavior, and digital finance has a more prominent role in the promotion of green innovation in large state-owned enterprises; digital finance indirectly promotes green innovation by improving the quality of enterprises' environmental in-formation disclosure and reducing financial constraints. Xue et al. (2022) [22] found that digital finance can promote the green innovation of enterprises in the heavy pollution industry, and the impact on the green innovation of heavy pollution enterprises in the maturity period is higher than that of enterprises in the growth period; digital finance promotes green innovation by alleviating corporate financing constraints and financial mismatches. Li et al. (2022) [23] found that the promotion effect of digital finance on enterprise green innovation persists and shows an upward trend over time, and with the increasing level of digital finance development, its impact on green innovation is more significant, and this effect is more obvious in state-owned enterprises, economically developed regions in the east, and high-pollution industries; digital finance can improve green innovation by reducing corporate financing constraints and improving the overall innovation capacity of cities. Rao et al. (2022) [24] found that digital finance can significantly increase the number of corporate green patent applications, the number of citations of green patents, and improve the quantity and quality of corporate green innovations, and this effect is stronger in eastern, state-owned, and mature firms, the development of digital finance can promote corporate green innovation by improving the transparency of enterprises, increasing the efficiency of inter-enterprise capital flows, and making the

allocation of financial resources more convenient. Li et al. (2023) [25] found that the promotion of digital finance on green innovation is mainly driven by the developmental drivers of the depth of use and level of digitization of digital finance; digital finance can promote corporate green innovation by improving the efficiency of financial services and alleviating capital misallocation. Li et al. (2023) [26] found that the effect of digital finance on green innovation is more obvious in state-owned enterprises and in regions with a lower degree of financial development and stronger financial regulation; digital finance promotes green innovation by alleviating capital constraints and increasing R&D investment. Ma et al. (2023) [27] found that digital finance and its coverage breadth, usage depth and digitization level can significantly improve the level of green innovation of enterprises; digital finance can improve the level of green technology innovation of enterprises by easing financing constraints and improving internal control.

In summary, the existing literature has studied the relationship between digital finance and corporate green innovation from different perspectives, and has achieved many valuable results, but there is still much room for expansion. First, the existing relevant literature has only explored the role mechanism of digital finance on corporate green innovation from the perspectives of alleviating financing constraints, improving corporate transparency, increasing R&D investment, enhancing financial flexibility, improving environmental information disclosure quality, mitigating financial mismatch, improving internal control, and solving internal and external information constraint, but lacks the examination of the role mechanism of digital finance in influencing corporate green innovation, such as stimulating consumer demand and alleviating distortion of the factor market. Second, the existing research on the heterogeneity of the impact of digital finance on enterprise green innovation is only based on the level of regional economic development, the degree of pollution in the industry, the nature of enterprise property rights, the life cycle of the enterprise, the level of regional financial development and the intensity of regional financial regulation, etc., and it lacks to analyze the heterogeneity of the impact of digital finance on enterprise green innovation from the perspective of the enterprise scale and whether it has the qualification of high and new technology. This study aims to address these deficiencies by analyzing the influence and mechanisms of digital finance on enterprise green innovation from the perspective of reducing information asymmetry, stimulating consumer demand and alleviating factor market distortion.

## 3. Theoretical analysis and research hypothesis

### 3.1 The direct impact of digital finance on enterprises' green innovation

To begin with, digital finance leverages its diverse range of financial services, such as mobile payment and online lending, to cater to the flexible and varied payment needs of different groups. This leads to an expansion of financing channels and methods accessible to enterprises, which, in turn, enhances the overall vitality of the financial market, and lays the foundation for financing green innovation projects of businesses [28]. Secondly, digital finance transcends the traditional limitations of the financial model by eliminating constraints of time and space. This allows customers to engage in all-day, contactless, and remote intelligent transactions [29]. It remarkably enhances the efficiency of capital flow between enterprises and promotes the transfer of financial resources towards high-quality green innovation projects. Lastly, digital finance uses its ability to analyze digital information from enterprises to provide differentiated financial services that cater to the unique needs of each business. This helps enterprises identify opportunities and risks during the process of green innovation, guiding them towards making more informed and reasonable decisions in this regard. Ultimately, this

approach facilitates finding the optimal path of green innovation practice for each enterprise [30]. Drawing from the preceding analysis, this paper proposes the following hypothesis:

**Hypothesis 1: The development of digital finance can promote green innovation of enterprises.**

## 3.2 Analysis of the mechanism of digital finance affecting enterprises' green innovation

**3.2.1 Mechanism of information asymmetry reduction.** Green innovation is the main source of the core competitiveness of modern enterprises and also determines the long-term development of enterprises; thus, its key information is generally not disclosed or shared with the public. Faced with significant information asymmetry, financial institutions are often dis-incentivized to provide credit for enterprise green innovation projects. Digital finance improves enterprise transparency, alleviates internal and external information asymmetry, and improves stakeholders' ability to use digital advantages to screen information [31]. When stakeholders possess comprehensive information regarding enterprise behavior, they are better equipped to recognize instances of "greenwashing," or even environmentally detrimental practices. This can help expose a company's environmental violations to some extent [32]. Upon receiving such signals, both the government and the public may issue mandatory or non-mandatory orders and denouncements aimed at achieving the dual goals of compulsory and social governance [33]. Financial institutions may also adjust their evaluation processes for enterprises, placing greater emphasis on environmental criteria. This can aid in the effective transfer of financial resources towards high-rated, environmentally responsible businesses. As a result, there will be less motivation for enterprises to engage in environmental speculation, and green innovation as a prosocial, sustainable development behavior will become increasingly valued by businesses.

**3.2.2 Mechanism of consumer demand stimulus.** Digital finance promotes various wealth management products to residents through the Internet platform, providing more choices for residents to invest and digital financial technology through the Internet, big data, and other means of assessing personal credit, so that residents can obtain more credit lines and better credit services, which is conducive to broadening the source of green consumption funds for residents [34, 35]. Digital finance provides consumers with more convenient means of payment, effectively reduces consumers' time costs and transaction costs, and improves consumers' green consumption shopping experience [36, 37]. At the same time, digital finance provides consumers with personalized green consumption financial services through the evaluation of residents' behavior data and the calculation of the carbon emissions reduced by users' green consumption behavior and guides residents to generate more demand for higher-level green products and green services, which in turn drives enterprises to accelerate innovation to improve product quality and service models [38].

**3.2.3 Mechanism of factor market distortion mitigation.** Based on advanced information technology, digital finance collects and analyzes the behaviors of different industries, different enterprises, and different individuals; conducts big data processing at a low cost; screens out low-quality green innovation projects; and then provides more support for high-quality green innovation projects, thereby reducing the distortion of factor markets, improving the green innovation environment of enterprises, and further achieving the purpose of "survival of the fittest" [39]. On the other hand, based on financial technology, digital finance activates funds scattered in the normal financial system, improves the circulation efficiency of funds, and alleviates the liquidity constraints of the capital factor market [40, 41]. Furthermore, the

development of digital finance can drive competition among banks and facilitate the gradual maturation of the financial system. This can alleviate the distortion of factor markets to some extent [42] and help enterprises overcome financing constraints, ultimately promoting their engagement in green innovation activities. As a result, the alleviation of factor market distortions has improved the overall green innovation environment for enterprises, increasing the vitality of more green innovation subjects and enhancing regional green innovation capabilities. This, in turn, has an "incentive effect" on improving the green innovation abilities of businesses in the region [43].

**3.2.4 Impact of intellectual property protection.**   Intellectual property protection plays a crucial role in safeguarding enterprises' green patents from being replicated or stolen, thereby enhancing their exclusivity and stimulating enthusiasm for green innovation [44]. A higher level of intellectual property protection also lowers the risk of infringement on enterprises' green innovation by leveraging digital finance. This, in turn, encourages businesses to continue pursuing green innovation and maintain their competitiveness in the market [45, 46]. On the contrary, the low level of intellectual property protection will increase the uncertainty of the transformation of green innovation achievements of enterprises, affect their competitive advantage, and thus weaken the positive incentive effect of digital finance on green innovation.

**3.2.5 Impact of environmental governance.**   Although the government's environmental governance behavior increases the production cost of enterprises, it also forces enterprises to conduct green innovation, thereby producing a compensatory effect [47]. Environmental governance behaviors within a certain intensity range release a signal to enterprises that green innovation is profitable, and enterprises have more motivation to use digital finance to raise funds to carry out green innovation [48], thereby reducing pollutant emissions and environmental resource consumption, improving product quality and market competitiveness, and offsetting the cost of environmental governance through the compensation effect of green innovation [49]. That is, compared with the situation of low environmental governance, the impact of digital finance on enterprise green innovation is more significant when the level of environmental governance is high. Drawing on the aforementioned analysis, this paper proposes the following assumptions:

**Hypothesis 2: Digital finance promotes green innovation of enterprises by reducing information asymmetry (Hypothesis 2a), stimulating consumer demand (Hypothesis 2b), and mitigating the distortion degree of factor markets (Hypothesis 2c).**

**Hypothesis 3: The improvement of intellectual property protection (Hypothesis 3a), environmental governance (Hypothesis 3b) strengthens the role of digital finance in promoting green innovation of enterprises.**

Based on the theoretical mechanisms and hypotheses outlined above, we have developed a conceptual framework depicted in Fig 1 Subsequently, in the following pages, we present the empirical tests conducted within this framework.

## 4. Research design

### 4.1 Data sources

The research sample for this study consisted of 2071 A-share listed companies in China between the years 2011 and 2021, alongside the corresponding prefecture-level city data where the enterprise is registered as the benchmark and matched the starting period (2011) measured by the digital financial index to construct a panel dataset from 2011 to 2021. In order to

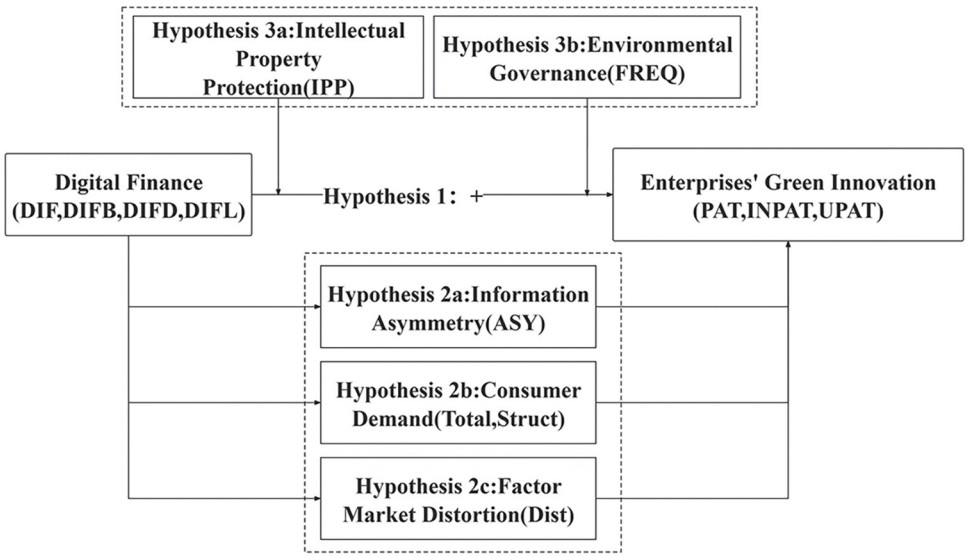

**Fig 1. Theoretical framework.**

improve the reliability and validity of the data, the sample was screened according to the following exclusion criteria: (1) companies listed during the sample period; (2) enterprises in the financial industry; (3) ST enterprises; (4) enterprises with serious data gaps. The continuous variables were also subjected to a 1% tailoring process, and 22781 "firm-year" observations were finally retained. Among them, the Peking University Digital Finance Index (2011–2021), compiled by the Digital Finance Research Center of Peking University, was the source of the prefecture-level digital finance index. Meanwhile, data on enterprise green patents were obtained from the CNRDS database, while financial data for the companies were sourced from the CSMAR database. Additionally, any incomplete data points were manually completed using the corresponding company's annual report.

## 4.2 Definition of variable

**4.2.1 Dependent variable.** Regarding enterprise green innovation, there is usually a one to two-year lag between the patent application and grant. However, since companies must engage in innovative practices to apply for patents, the number of patent applications serves as a more reliable indicator than the number of granted patents. Drawing on Wu et al.'s study [50], to measure enterprise green innovation, this study employed three indicators: *PAT*, *INPAT*, and *UPAT*, and the metrics align with the quantity of green patent, green invention patent, and green utility model patent applications submitted by listed companies. Additionally, the study conducted a robustness test using the number of cited green patents applied by enterprises (*GPR*) as an alternative indicator of enterprise green innovation.

**4.2.2 Independent variable.** Digital finance (*DIF*). To measure digital finance, this study followed the methodology established in previous research by utilizing the total digital finance index and its three sub-dimensions, namely digital financial coverage breadth (*DIFB*), digital financial usage depth (*DIFD*), and digital financial digitization level (*DIFL*). These proxy variables were compiled by the Peking University Digital Finance Research Center and Ant Financial Services Group. Furthermore, this study used the first-order lag term and the second-order lagging term of digital finance as their alternative indicators in the robustness test.

### 4.2.3 Mechanism variables.

1. *Information asymmetry (ASY)*. Drawing on Bharath et al.'s [51] idea, this study established a comprehensive indicator to measure the degree of information asymmetry between firms and the external capital market by conducting a multidimensional analysis of the trading information of individual shares of firms. Due to the serious lack of high-frequency trading data in China's securities market, this study used Amihud et al.'s [52] liquidity ratio indicator (*LR*), Amihud's [53] illiquidity ratio indicator (*ILL*), and Pastor et al.'s [54] return inversion indicator (*GMA*) to construct a composite indicator by extracting the first principal component to measure the degree of external market asymmetry of firms. The measures of each indicator are shown below:

$$LR_{it} = -\frac{1}{D_{it}} \sum\nolimits_{k=1}^{D_{it}} \sqrt{\frac{V_{it}(k)}{|r_{it}(k)|}} \qquad (1)$$

$$ILL_{it} = -\frac{1}{D_{it}} \sum\nolimits_{k=1}^{D_{it}} \sqrt{\frac{|r_{it}(k)|}{V_{it}(k)}} \qquad (2)$$

In the calculation of *LR* and *ILL*, $rit(k)$ denotes the stock return of firm i on the kth trading day in year t, $Vit(k)$ denotes the daily volume, and $Dit$ denotes the number of trading days of firm i's stock in year t. The calculation procedure for measuring liquidity based on yield inversion is as follows:

$$r_{it}^e(k) = \theta it + \varphi itrit(k-1) + \gamma itVit(k-1)sign[r_{it}^e(k-1)] + \varepsilon it(k) \qquad (3)$$

In Eq (3), $r_{it}^e(k)$ is the excess return, which is calculated using the market return weighted by the outstanding market capitalization, and the coefficient $\gamma it$ estimated in Eq (3)) is the yield inversion indicator (*GMA*). When the degree of information asymmetry is high, the larger the *LR*, *ILL*, and *GMA* indicators, the larger the index of the degree of information asymmetry (*ASY*) constructed by extracting the first principal component.

2. *Consumer demand (Consume)*. The increase in consumer demand is manifested in two specific aspects: first, the horizontal effect, that is, the increase in total consumption (*Total*), measured by the per capita consumption expenditure of residents, and second the structural effect, that is, the optimization and upgrading of the consumption structure (*Struct*), measured by the per capita disposable income of residents.

3. *Distortion in factor markets (Dist)*. Due to the lack of product prices and factor input volumes in previous years, measuring the level of factor market distortion in different regions of China presents a complex challenge. Drawing on the study of Kong et al. [55], this study constructed the factor market distortion indicators by using the marketization indices of the overall market, product market, and factor market in Fan Gang et al.'s "China Marketization Process Index Report" [56] and adopted the relative difference between the factor marketization level of each province and the highest factor marketization level in the sample as the proxy variable of factor market distortion. Specifically, the factor market distortion indicator constructed in this study is:

$$Dist_{it} = [\max(fmarket_{it}) - fmarket_{it}] / \max(fmarket_{it}) \qquad (4)$$

where fmarketit is the index of the degree of regional factor market development.

Obviously, the factor distortion index we constructed not only reflects the relative differences in the degree of factor market distortion between regions but also responds to the changes of regional factor market distortion itself over time.

4. *Intellectual property protection (IPP)*. Referring to the research of Park (2008) [57], the patent infringement rate as the proxy variable of intellectual property protection was taken as equal to "1- patent infringement rate", in which the formula employed in this study determines the rate of accepted patent infringement disputes per year by a provincial intellectual property office, expressed as a fraction of the total number of patents granted in that province until that point, and then subtracts that fraction from one. The greater the rate of patent non-infringement, the better the effect of regional intellectual property protection. The number of patent infringement disputes comes from the website of China Intellectual Property Protection Bureau.

5. *Environmental governance (FREQ)*. Drawing on Chen et al. (2016) [58], this study used the frequency of words related to the term "environmental protection" in municipal government work reports (specifically: environmental protection, pollution, energy consumption, emission reduction, emissions, ecology, green, low-carbon, air, chemical oxygen demand, sulfur dioxide, carbon dioxide, PM10, and PM2.5, etc.) as a proxy for environmental governance. The government work report serves as a comprehensive guide for the government's initiatives and actions, as it is an outline for the administration and implementation of the decisions and resolutions of the authorities in accordance with the law. Therefore, a thorough analysis of the frequency and proportion of environment-related terms used in the government work report can provide a more comprehensive understanding of the effectiveness of the government's environmental governance policies.

**4.2.4 Control variables.** To avoid the estimation bias caused by missing variables as much as possible, according to the previous theoretical discussion and existing literature research, in this study, various micro-level variables were taken into consideration as control variables for enterprises, such as the size of the enterprise, age of the enterprise, asset-liability ratio, proportion of shares held by the largest shareholder, management expense ratio, sales revenue growth rate, return on equity, Tobin's Q value, management shareholding ratio, board of directors' independence, dual role of the board chairman, nature of property, and board size. The specific definitions of the variables involved in the model are presented in **Table 1**.

## 4.3 Model setting

Considering the truncation feature of the green patent variable in the sample, some patent data in the sample are zero and relatively concentrated, which may lead to selective bias in the research sample. Drawing on the research of Estrada et al. [59], the Tobit model was utilized in this study for estimation, with the following settings:

$$Y_{it} = \alpha + \beta DIF_{it} + \sum \varphi CV_{it} + \varepsilon_{it} \tag{5}$$

Among them, the dependent variable *Y* is enterprise green innovation, with enterprise green innovation (*PAT*), enterprise green invention innovation (*INPAT*), and enterprise green utility model innovation (*UPAT*) as proxy variables. The core independent variable is the digital financial index (*DIF*) of prefecture-level cities, and *CV* includes the aforementioned control variables. $\varepsilon_{it}$ represents the random error term of the model.

**Table 1. Variable definitions.**

| Type | Symbol | Variable | Description |
|---|---|---|---|
| Dependent Variable | PAT | Enterprise green innovation | The quantity of green patent applications of listed companies |
| | INPAT | Enterprise green invention innovation | The quantity of green invention patent applications of listed companies |
| | UPAT | Enterprise green utility model innovation | The quantity of green utility model patent applications of listed companies |
| Independent Variable | DIF | Digital finance | Peking University Digital Finance Index (2011–2021) |
| | DIFB | Digital financial coverage breadth | Peking University Digital Finance Index (2011–2021) |
| | DIFD | Digital financial usage depth | Peking University Digital Finance Index (2011–2021) |
| | DIFL | Digital financial digitization level | Peking University Digital Finance Index (2011–2021) |
| Mechanism Variables | ASY | Information asymmetry | Using daily trading data to calculate |
| | Total | Consumption total | Per capita consumption expenditure of residents |
| | Struct | Consumption structure | Per capita disposable income of residents |
| | Dist | Distortion in the factor market | The gap between the development level of factor markets in each province and the maximum value |
| | IPP | Intellectual property protection | 1- Number of patent infringement cases accepted/number of patents granted by the province in that year. |
| | FREQ | Environmental governance | The ratio of words pertaining to 'environmental protection' in the local government's work report to the overall word count of the report. |
| Control Variables | Size | Enterprise size | ln (total assets) |
| | Age | Enterprise age | ln (sample year—company listing year + 1) |
| | lev | Asset–liability ratio | Total liabilities/total assets |
| | Top1 | Proportion of shares held by the largest shareholder | Shareholding percentage of the largest shareholder/total share capital |
| | Mfee | Management expense ratio | Current management expenses of the enterprise / operating income |
| | Growth | Sales revenue growth rate | Current operating income/previous operating income |
| | ROE | Return on equity | After-tax profit of the company/net assets |
| | TobinQ | Tobin's Q value | Market value / (total assets—net intangible assets—net goodwill) |
| | Mshare | Management shareholding ratio | Management shareholding amount/total share capital |
| | Indep | Independence of the board of directors | Number of independent directors/numbers of board members |
| | Dual | Dual role of the board chairman | If the chairman and the general manager are the same person, assign a value of 1, otherwise 0. |
| | SOE | Nature of property | It is represented using 0–1 dummy variables, where state-owned = 1 and non-state-owned = 0. |
| | Board | Board size | ln (total number of members in the company's board of directors) |

## 4.4 Descriptive statistics

The descriptive statistics of the variables are displayed in Table 2 Specifically, *PAT* has a mean of 8.318, with a standard deviation of 48.44, a minimum value of 0, and a maximum value of 1612, indicating significant variations in green innovation among Chinese enterprises. The skewness of *PAT* is greater than 18, showing a typical right-skewness distribution, and the kurtosis is greater than 441, indicating that the distribution is peak-like compared with the normal distribution. *INPAT* and *UPAT* show the same characteristics. In addition, *DIF* has a mean of 213.1, a standard deviation of 77.12 a minimum value of 21.26, and a maximum value of 359.7, suggesting that the level of digital finance development across regions in China is relatively uneven, showing polarization. The skewness of *DIF* is close to 0, and the kurtosis is less than 3, indicating that the distribution of *DIF* is flat compared with the normal distribution without obvious skewness. Its coverage breadth, usage depth, and digitization level show the same characteristics.

**Table 2. Descriptive statistics.**

| Variable | N | Mean | Std. Dev. | Min | Median | Max | Skewness | Kurtosis |
|---|---|---|---|---|---|---|---|---|
| PAT | 22781 | 8.318 | 48.44 | 0 | 0 | 1612 | 18.31 | 441.4 |
| INPAT | 22781 | 4.876 | 33.53 | 0 | 0 | 1381 | 21.43 | 605.1 |
| UPAT | 22781 | 3.442 | 19.02 | 0 | 0 | 709 | 19.09 | 488.9 |
| DIF | 22781 | 213.1 | 77.12 | 21.26 | 222.2 | 359.7 | -0.313 | 2.189 |
| DIFB | 22781 | 213.2 | 76.08 | -10.49 | 219.8 | 371.8 | -0.212 | 2.343 |
| DIFD | 22781 | 208.5 | 78.32 | 12.49 | 216.1 | 354.3 | -0.220 | 1.994 |
| DIFL | 22781 | 221.3 | 90.41 | 3.390 | 246.8 | 581.2 | -0.670 | 2.313 |
| Size | 22781 | 22.39 | 1.398 | 12.24 | 22.23 | 48.31 | 1.182 | 14.01 |
| Age | 22781 | 2.914 | 0.350 | 0.693 | 2.996 | 3.829 | -1.086 | 4.968 |
| lev | 22781 | 0.418 | 0.254 | -1.855 | 0.421 | 9.429 | 2.648 | 83.88 |
| Top1 | 22781 | 0.0920 | 0.166 | 0.009 | 0.009 | 0.961 | 1.985 | 6.055 |
| Mfee | 22781 | 0.211 | 14.02 | 0 | 0.0730 | 2115 | 150.5 | 22700 |
| Growth | 22781 | -0.298 | 0.903 | -13.39 | -0.129 | 57.41 | 15.77 | 824.7 |
| ROE | 22781 | 0.0490 | 0.231 | -7.016 | 0.0640 | 14.02 | 7.325 | 702.6 |
| TobinQ | 22781 | 0.213 | 0.838 | 0 | 0.005 | 25.51 | 8.657 | 132.0 |
| Mshare | 22781 | 0.00800 | 0.0480 | 0 | 0 | 0.692 | 7.737 | 70.38 |
| Indep | 22781 | 0.375 | 0.0560 | 0 | 0.354 | 0.810 | 1.250 | 6.493 |
| Dual | 22781 | 0.232 | 0.422 | 0 | 0 | 1 | 1.268 | 2.607 |
| SOE | 22781 | 0.430 | 0.495 | 0 | 0 | 1 | 0.284 | 1.081 |
| Board | 22781 | 2.141 | 0.203 | 0 | 2.197 | 3.819 | -0.338 | 5.310 |

## 5. Analysis of the results of the return

### 5.1 Benchmark regression results

Table 3 displays the outcomes of the benchmark regression analysis conducted to examine the association between digital finance and green innovation in enterprises. In models (1) to (3), only the "time–province" fixed effects were controlled, and the results showed that the impact of digital finance development (DIF) on enterprise green innovation was examined through a regression analysis, which revealed positive coefficients ($\beta PAT$ = 0.2412, $\beta INPAT$ = 0.2697, $\beta UPAT$ = 0.1054) for the total level of green innovation (PAT), green invention innovation (INPAT), and green utility model innovation (UPAT). All coefficients were statistically significant at the 1% level. After including the relevant control variable set (models (4) to (6)), the statistical significance level of digital finance development (DIF) on the total green innovation level of enterprises (PAT) and green utility model innovation (UPAT) regression decreased. The research results show that under the influence of digital finance, enterprises' green innovation ability is gradually strengthened. With the support of digital finance, enterprises have improved their ability to collect, integrate and analyze information, which can help enterprises judge the status of green innovation and market potential, and improve the effectiveness of green innovation decisions of enterprises. In addition, under the supervision pressure from outside the market, enterprises will pay more attention to how to improve the core innovation competitiveness, so as to concentrate resources on such green invention innovation activities with high gold content, and the promotion effect is weak for those patent innovations with low economic potential. The abovementioned results prove Hypothesis 1.

In order to provide a more precise representation of the impact of digital finance on enterprise green innovation, this research has classified the digital finance index into three distinct levels: the coverage breadth, usage depth and digitization level. Based on this, this study

**Table 3. The impact of digital finance on enterprise green innovation: Benchmark regression.**

|  | (1) | (2) | (3) | (4) | (5) | (6) |
|---|---|---|---|---|---|---|
|  | *PAT* | *INPAT* | *UPAT* | *PAT* | *INPAT* | *UPAT* |
| *DIF* | 0.2412*** | 0.2697*** | 0.1054*** | 0.1412** | 0.2053*** | 0.0486 |
|  | (3.3948) | (3.7023) | (2.9687) | (2.2559) | (3.2001) | (1.5843) |
| *Size* |  |  |  | 19.0160*** | 15.5699*** | 8.2557*** |
|  |  |  |  | (4.9287) | (4.2377) | (5.3600) |
| *Age* |  |  |  | -29.6241*** | -20.7352*** | -14.6130*** |
|  |  |  |  | (-3.7899) | (-3.6727) | (-3.6221) |
| *lev* |  |  |  | -2.3339 | -8.6100 | 7.4236* |
|  |  |  |  | (-0.2669) | (-1.1429) | (1.9093) |
| *Top1* |  |  |  | 6.6467 | -8.9407 | 15.4544** |
|  |  |  |  | (0.3963) | (-0.6518) | (2.0009) |
| *Mfee* |  |  |  | -12.1058** | -5.2157 | -8.1610** |
|  |  |  |  | (-2.2065) | (-1.5865) | (-2.3993) |
| *Growth* |  |  |  | 0.5067 | 0.6769 | -0.3611 |
|  |  |  |  | (0.4767) | (0.7370) | (-0.6745) |
| *ROE* |  |  |  | 5.6204 | 4.2247 | 4.6433** |
|  |  |  |  | (1.5890) | (1.4180) | (2.4021) |
| *TobinQ* |  |  |  | 3.3534 | 2.3588 | 1.1433 |
|  |  |  |  | (1.0608) | (1.1141) | (0.5866) |
| *Mshare* |  |  |  | -1.9191 | 7.9416 | -5.1134 |
|  |  |  |  | (-0.1018) | (0.5649) | (-0.4311) |
| *Indep* |  |  |  | 43.2382 | 27.8085 | 23.7394 |
|  |  |  |  | (1.4495) | (1.3230) | (1.4632) |
| *Dual* |  |  |  | 4.8229* | 4.6548** | 1.1462 |
|  |  |  |  | (1.7508) | (2.1169) | (0.8807) |
| *SOE* |  |  |  | 1.5075 | 2.7335 | -0.7720 |
|  |  |  |  | (0.5505) | (1.1431) | (-0.6119) |
| *Board* |  |  |  | 4.5568 | 8.3589 | -1.8162 |
|  |  |  |  | (0.5237) | (1.0525) | (-0.5076) |
| *year* | Yes | Yes | Yes | Yes | Yes | Yes |
| *province* | Yes | Yes | Yes | Yes | Yes | Yes |
| N | 22781 | 22781 | 22781 | 22781 | 22781 | 22781 |
| PseudoR$^2$ | 0.0128 | 0.0141 | 0.0190 | 0.0310 | 0.0347 | 0.0451 |

Notes: All t-statistics are presented in parentheses under the estimated coefficient. ***, **, and * indicate 1%, 5% and 10% of significance levels, respectively.

analyzed which dimensions of digital finance development can significantly promote enterprise green innovation. The results in Table 4 show the impact of the development of the dimension of "coverage breadth—usage depth—digitization level" on enterprises' green innovation activities: the regression coefficient of digital financial coverage breadth (*DIFB*) on enterprises' green invention and innovation ($\beta INPAT = 0.1549$) is positive, passing the 1% statistical significance test. The regression coefficients of total green innovation level and green utility model innovation ($\beta PAT = 0.1136$ and $\beta UPAT = 0.0439$) were positive, and the significance decreased. The usage depth of digital finance (*DIFD*) only had a significant effect on green invention innovation ($\beta INPAT = 0.1567$). The digitization level of digital finance (*DIFL*) has no significant impact on enterprises' green innovation. This shows that the development of digital finance mainly stimulates the green innovation of enterprises by expanding the

**Table 4. The impact of digital finance development on enterprise green innovation: Indicator dimensionality reduction.**

|  | (1) | (2) | (3) |
|---|---|---|---|
|  | *PAT* | *INPAT* | *UPAT* |
| *DIFB* | 0.1136** | 0.1549*** | 0.0439* |
|  | (2.3816) | (3.1593) | (1.8865) |
| *Control* | Yes | Yes | Yes |
| *year* | Yes | Yes | Yes |
| *province* | Yes | Yes | Yes |
| N | 22781 | 22781 | 22781 |
| PseudoR$^2$ | 0.0311 | 0.0347 | 0.0452 |
| *DIFD* | 0.0917 | 0.1567*** | 0.0242 |
|  | (1.5471) | (2.7426) | (0.8178) |
| *Control* | Yes | Yes | Yes |
| *year* | Yes | Yes | Yes |
| *province* | Yes | Yes | Yes |
| N | 22781 | 22781 | 22781 |
| PseudoR$^2$ | 0.0310 | 0.0345 | 0.0451 |
| *DIFL* | 0.0071 | 0.0354 | -0.0136 |
|  | (0.2329) | (1.3041) | (-0.9605) |
| *Control* | Yes | Yes | Yes |
| *year* | Yes | Yes | Yes |
| *province* | Yes | Yes | Yes |
| N | 22781 | 22781 | 22781 |
| PseudoR$^2$ | 0.0309 | 0.0343 | 0.0451 |

Notes: All t-statistics are presented in parentheses under the estimated coefficient. ***, **, and * indicate 1%, 5% and 10% of significance levels, respectively.

coverage and deep mining. The coverage breadth of digital finance can better reflect the fairness of digital finance, so that small and medium-sized enterprises can reach financial services, and reduce the uneven allocation of financial resources. The usage depth of digital finance reflects the application results of digital finance, and describes the specific financial functions of digital finance in the business activities of enterprises. Digitization level of digital finance, as the embodiment of the low threshold and low cost characteristics of digital finance, can increase the demand for financial services, but compared with the breadth and depth of digital financial applications, its green innovation effect is relatively small. If the development of digital finance only relies on digitalization without achieving extensive coverage and deep mining, it is difficult to provide support for microeconomic entities, nor can it provide sustained impetus for high-quality economic development.

## 5.2 Endogenous analysis

Our estimation model may also face the challenge of reverse causation: in regions where there is a significant emphasis on enterprise green innovation, it is observed that there exists a positive correlation between the quality of economic conditions and institutional environment, which may provide a more favorable environment for the advancement of digital finance. Therefore, the correlation between the error perturbation term and the core independent variables in the model has not been completely eradicated. With this in mind, we further use instrumental variable models to verify whether the development of digital finance affects

**Table 5. Regression results of instrumental variables: Provincial distance.**

| | (1) | (2) | (3) |
|---|---|---|---|
| | IV Tobit Marginality | Tobit Marginality | Tobit Marginality |
| *DIF* | 0.1180** | | 0.0584*** |
| | (2.0716) | | (9.9834) |
| *distance* | | -0.0079** | -0.0040 |
| | | (-2.1161) | (-1.0590) |
| *Control* | Yes | Yes | Yes |
| *distance* | -0.0664*** | | |
| | (-9.8601) | | |
| *Control* | Yes | Yes | Yes |
| N | 22781 | 22781 | 22781 |
| F | 21.92 | | |
| PseudoR$^2$ | | 0.0365 | 0.0398 |

Notes: All t-statistics are presented in parentheses under the estimated coefficient. ***, **, and * indicate 1%, 5% and 10% of significance levels, respectively.

enterprise green innovation in a causal sense. Specifically, drawing on the methods of Yu et al. (2019) [60], this study used the IV Tobit estimation method to examine the impact of digital financial on enterprise green innovation. Drawing on the practice of Zhang et al. (2020) [61], this study took the spherical distance from the enterprise to the provincial capital (*distance*) as an instrumental variable. A provincial capital is usually the economic center of a province and should also be a center for digital financial development; the closer it is to the provincial capital, the better the development of digital finance, and the instrumental variable is obviously correlated with the degree of digital finance development in the region.

The estimation results in Column (1) of Table 5 were obtained using the IV Tobit model with an instrumental variable. The *DIF* coefficient was found to be statistically significant at the 5% level, indicating that the conclusion that "digital finance promotes green innovation in enterprises" still holds. Using the Anderson-Rubin weak instrumental variable test, our study rejected the null hypothesis that the regression coefficient of endogenous variables is equal to 0 with high statistical significance. This suggests a strong correlation between the instrumental variable and the variable *DIF*, which was constructed for this study. In addition, this study estimated the equation using the two-stage least-squares (2SLS) method to obtain the first stage Cragg–Donald Wald F statistic, and the results show that the Cragg–Donald Wald F statistic corresponding to the instrumental variable is 21.92, in which the critical value at the 10% significance level was surpassed, leading us to reject the null hypothesis that the instrumental variable is weak. In this study, the exogenous nature of instrumental variables was preliminarily tested: firstly, the potential endogenous variable *DIF* in Eq (1) was replaced with the instrumental variable, in which we directly tested the impact of the instrumental variable on the green innovation of enterprises, and the results presented in Column (2) of Table 5 indicate that the coefficient of the instrumental variable is significantly negative. Then, the variable *DIF* was added to the equation, the impact of the variable DIF and the instrumental variable on enterprise green innovation was compared, and the results of Column (3) of Table 5 show that the coefficient of the instrumental variable is negative but not significant, while the variable *DIF* coefficient is significantly positive, which indicates that the explanatory force of the instrumental variable on the model is absorbed by the variable *DIF*, and the instrumental variable indirectly affects the green innovation of enterprises through the variable *DIF*. The instrumental variable constructed in this study is reasonable. Collectively, after conducting correlation

tests, the selected instrumental variables in this study are confirmed to be valid. Further regression analysis demonstrates that the influence of digital finance on promoting green innovation among enterprises remains significant even after accounting for potential endogeneity issues, which also proves the robustness of the previous conclusion.

### 5.3 Robustness test

**5.3.1 Addition of control variables.**   This study incorporated macro-level data on various factors, including the share of secondary industry value added in GDP (GDP2), expenditure on science and education as a percentage of total regional fiscal expenditure (sciedugdp), foreign direct investment as a percentage of GDP (FDI), government subsidies received by enterprises (sub), and marketization index of prefecture-level cities (Market). These factors were added to the model, and the regression results are presented in Columns (1)—(3) of Table 6. The findings indicate that the regression coefficient for digital finance retains its statistical significance with a positive value, indicating that the benchmark regression results are generally robust.

**5.3.2 Replacement regression model.**   To conduct the analysis, a bidirectional fixed-effect model was utilized. The regression results are presented in Columns (4)-(6) of Table 6, and the regression coefficient for digital finance remains significantly positive. This finding is consistent with the previous conclusion.

**5.3.3 Exclusion of some data.**   On the one hand, it is noteworthy that the 2015 Chinese stock market crash may have had an impact on both the development of digital finance and the green innovation behavior of enterprises, and this study excluded the 2015 data; on the other hand, due to the large economic specificity of the municipalities in China, it is also possible that there are differences in the development of digital finance and the green innovation activities among enterprises., thus, this study excluded the sample data of the municipalities and re-ran the regression. The regression results are shown in Columns (7)—(12) of Table 6. The positive statistical significance of the regression coefficient for digital finance persists, and the results are still robust.

**5.3.4 Replacement of the independent variable.**   The value of a green patent and the quality of green innovation for an enterprise that has applied for the patent may be positively correlated with the frequency of citations of the patent. To evaluate the quality of green innovation among enterprises, this study employed the number of green patents cited (*GPR*) as a measure. Additionally, the research examined how digital finance and its sub-dimensions influence ecological innovation in enterprises. The regression analysis results are presented in Columns (13)—(16) of Table 6, and they confirm that the positive effect of digital finance on green innovation is still significant. This further supports the robustness of the benchmark regression results.

## 6. Heterogeneity analysis

### 6.1 Sub-sample study based on firm size

This study divided the sample of enterprises according to their total assets (enterprises with total assets above the mean are large enterprises; otherwise, they are SMEs) and examined the variability of the impact of digital finance on enterprises of different sizes. The results, presented in Table 7, demonstrate the significantly positive impact of both digital finance and its coverage breadth and usage depth indicators on SMEs' green innovation, green invention innovation and green utility model innovation of enterprises, the regression coefficients for digital financial and digitization level indicators are significantly positive only in relation to green innovation and green invention innovation among SMEs. However, the regression

**Table 6. Robustness test results.**

| | Increased control variables | | | Replacement Regression model | | |
|---|---|---|---|---|---|---|
| | **(1)** | **(2)** | **(3)** | **(4)** | **(5)** | **(6)** |
| | **PAT** | **INPAT** | **UPAT** | **PAT** | **INPAT** | **UPAT** |
| *DIF* | 0.1513** | 0.2130*** | 0.0483 | 0.0778*** | 0.0653*** | 0.0126 |
| | (2.5074) | (3.5005) | (1.5579) | (2.7056) | (2.8491) | (1.2562) |
| *Control* | Yes | Yes | Yes | Yes | Yes | Yes |
| *year* | Yes | Yes | Yes | Yes | Yes | Yes |
| *province* | Yes | Yes | Yes | Yes | Yes | Yes |
| N | 22781 | 22781 | 22781 | 22781 | 22781 | 22781 |
| PseudoR$^2$ | 0.0354 | 0.0398 | 0.0476 | | | |
| | Excluding 2015 | | | Excluding municipalities | | |
| | **(7)** | **(8)** | **(9)** | **(10)** | **(11)** | **(12)** |
| | **PAT** | **INPAT** | **UPAT** | **PAT** | **INPAT** | **UPAT** |
| *DIF* | 0.1559** | 0.2134*** | 0.0612** | 0.1306** | 0.1592*** | 0.0469** |
| | (2.4811) | (3.3093) | (2.0141) | (2.5627) | (3.1375) | (2.0733) |
| *Control* | Yes | Yes | Yes | Yes | Yes | Yes |
| *year* | Yes | Yes | Yes | Yes | Yes | Yes |
| *province* | Yes | Yes | Yes | Yes | Yes | Yes |
| N | 20710 | 20710 | 20710 | 18271 | 18271 | 18271 |
| PseudoR$^2$ | 0.0315 | 0.0353 | 0.0463 | 0.0306 | 0.0351 | 0.0440 |
| | Replaced Independent variable | | | | | |
| | **(13)** | **(14)** | **(15)** | **(16)** | | |
| | **GPR** | **GPR** | **GPR** | **GPR** | | |
| *DIF* | 1.5224** | | | | | |
| | (2.0931) | | | | | |
| *DIFB* | | 1.1562** | | | | |
| | | (2.0790) | | | | |
| *DIFD* | | | 1.1198* | | | |
| | | | (1.7807) | | | |
| *DIFL* | | | | 0.3041 | | |
| | | | | (1.1499) | | |
| *Control* | Yes | Yes | Yes | Yes | | |
| *year* | Yes | Yes | Yes | Yes | | |
| *province* | Yes | Yes | Yes | Yes | | |
| N | 22781 | 22781 | 22781 | 22781 | | |
| PseudoR$^2$ | 0.0169 | 0.0169 | 0.0168 | 0.0168 | | |

Notes: All t-statistics are presented in parentheses under the estimated coefficient. ***, **, and * indicate 1%, 5% and 10% of significance levels, respectively.

coefficients of digital finance and its sub-dimension indicators do not show significant effects on green innovation among large enterprises.

This may be because SMEs are more likely to be excluded from the threshold of traditional financial services due to their shortage of mortgage resources, high operational risks, and relatively imperfect credit records and have very limited financing channels, while due to the inclusiveness of digital finance, SMEs are more motivated to use them to obtain green innovation financing. However, large enterprises have relatively sufficient funds and face fewer financing constraints, and digital finance has less impact on them.

**Table 7. Sample results of enterprise size.**

| | SMEs | | | Big enterprises | | |
|---|---|---|---|---|---|---|
| | (1) | (2) | (3) | (4) | (5) | (6) |
| | *PAT* | *INPAT* | *UPAT* | *PAT* | *INPAT* | *UPAT* |
| *DIF* | 0.0900*** | 0.0993*** | 0.0450*** | 0.0774 | 0.1074 | 0.0005 |
| | (3.9953) | (5.2167) | (2.6851) | (0.6532) | (1.0253) | (0.0094) |
| *Control* | Yes | Yes | Yes | Yes | Yes | Yes |
| *year* | Yes | Yes | Yes | Yes | Yes | Yes |
| *province* | Yes | Yes | Yes | Yes | Yes | Yes |
| N | 12397 | 12397 | 12397 | 10384 | 10384 | 10384 |
| PseudoR$^2$ | 0.0380 | 0.0412 | 0.0444 | 0.0249 | 0.0262 | 0.0369 |
| *DIFB* | 0.0658*** | 0.0714*** | 0.0341*** | 0.0705 | 0.0848 | 0.0115 |
| | (3.9989) | (5.1803) | (2.8310) | (0.7697) | (1.0530) | (0.2711) |
| *Control* | Yes | Yes | Yes | Yes | Yes | Yes |
| *year* | Yes | Yes | Yes | Yes | Yes | Yes |
| *province* | Yes | Yes | Yes | Yes | Yes | Yes |
| N | 12397 | 12397 | 12397 | 10384 | 10384 | 10384 |
| PseudoR$^2$ | 0.0380 | 0.0412 | 0.0445 | 0.0249 | 0.0262 | 0.0369 |
| *DIFD* | 0.0689*** | 0.0816*** | 0.0312* | 0.0442 | 0.0770 | -0.0125 |
| | (3.0801) | (4.4065) | (1.7665) | (0.4073) | (0.8122) | (-0.2483) |
| *Control* | Yes | Yes | Yes | Yes | Yes | Yes |
| *year* | Yes | Yes | Yes | Yes | Yes | Yes |
| *province* | Yes | Yes | Yes | Yes | Yes | Yes |
| N | 12397 | 12397 | 12397 | 10384 | 10384 | 10384 |
| PseudoR$^2$ | 0.0372 | 0.0398 | 0.0439 | 0.0249 | 0.0262 | 0.0369 |
| *DIFL* | 0.0229** | 0.0264*** | 0.0072 | -0.0159 | 0.0126 | -0.0293 |
| | (2.3163) | (2.9670) | (0.9498) | (-0.3085) | (0.2822) | (-1.2678) |
| *Control* | Yes | Yes | Yes | Yes | Yes | Yes |
| *year* | Yes | Yes | Yes | Yes | Yes | Yes |
| *province* | Yes | Yes | Yes | Yes | Yes | Yes |
| N | 12397 | 12397 | 12397 | 10384 | 10384 | 10384 |
| PseudoR$^2$ | 0.0367 | 0.0383 | 0.0436 | 0.0249 | 0.0262 | 0.0369 |

Notes: All t-statistics are presented in parentheses under the estimated coefficient. ***, **, and * indicate 1%, 5% and 10% of significance levels, respectively.

## 6.2 Sub-sample research based on high-tech/non-high-tech industries

The study categorized enterprises into high-tech and non-high-tech industries by referring to the high-tech enterprise recognition announcement and re-examination announcement published in the WIND database. Table 8 displays the regression outcomes of the two subgroups. The results indicate that the digital finance and its coverage breadth, usage depth and digitalization level indicators have a significantly positive effect on green innovation, green invention innovation, and green utility model innovation of high-tech industry enterprises. Conversely, the regression analyses fail to confirm the significance of the impact of digital finance and its sub-dimensional indicators on green innovation of non-high-tech industry enterprises.

This may be because compared with non-high-tech enterprises, high-tech enterprises have stronger motivation to finance green innovation projects due to their knowledge-intensive and environment-friendly characteristics. Specifically, green innovation projects account for a large proportion of the income of high-tech enterprises, and green innovation research and development itself is characterized by high investment, high risk and long duration, which

**Table 8. Sample results of high-tech/non-high-tech industries.**

| | Non-high-tech industries | | | High-tech industries | | |
|---|---|---|---|---|---|---|
| | (1) | (2) | (3) | (4) | (5) | (6) |
| | PAT | INPAT | UPAT | PAT | INPAT | UPAT |
| DIF | -0.0876 | -0.0003 | -0.0544 | 0.3459*** | 0.3703*** | 0.1219*** |
| | (-0.9011) | (-0.0034) | (-1.0374) | (4.1251) | (3.6239) | (5.3596) |
| Control | Yes | Yes | Yes | Yes | Yes | Yes |
| year | Yes | Yes | Yes | Yes | Yes | Yes |
| province | Yes | Yes | Yes | Yes | Yes | Yes |
| N | 13816 | 13816 | 13816 | 8965 | 8965 | 8965 |
| PseudoR$^2$ | 0.0384 | 0.0432 | 0.0540 | 0.0411 | 0.0429 | 0.0662 |
| DIFB | -0.0567 | 0.0058 | -0.0351 | 0.2442*** | 0.2563*** | 0.0906*** |
| | (-0.7652) | (0.0929) | (-0.8731) | (4.2962) | (3.7491) | (5.3653) |
| Control | Yes | Yes | Yes | Yes | Yes | Yes |
| year | Yes | Yes | Yes | Yes | Yes | Yes |
| province | Yes | Yes | Yes | Yes | Yes | Yes |
| N | 13816 | 13816 | 13816 | 8965 | 8965 | 8965 |
| PseudoR$^2$ | 0.0384 | 0.0432 | 0.0540 | 0.0411 | 0.0427 | 0.0665 |
| DIFD | -0.0851 | 0.0059 | -0.0524 | 0.2707*** | 0.2973*** | 0.0860*** |
| | (-0.8775) | (0.0726) | (-1.0321) | (3.5257) | (3.2705) | (4.0548) |
| Control | Yes | Yes | Yes | Yes | Yes | Yes |
| year | Yes | Yes | Yes | Yes | Yes | Yes |
| province | Yes | Yes | Yes | Yes | Yes | Yes |
| N | 13816 | 13816 | 13816 | 8965 | 8965 | 8965 |
| PseudoR$^2$ | 0.0384 | 0.0432 | 0.0540 | 0.0399 | 0.0412 | 0.0647 |
| DIFL | -0.0392 | -0.0317 | -0.0245 | 0.1129** | 0.1454** | 0.0209 |
| | (-0.8369) | (-0.7504) | (-1.0401) | (2.1781) | (2.4444) | (1.5637) |
| Control | Yes | Yes | Yes | Yes | Yes | Yes |
| year | Yes | Yes | Yes | Yes | Yes | Yes |
| province | Yes | Yes | Yes | Yes | Yes | Yes |
| N | 13816 | 13816 | 13816 | 8965 | 8965 | 8965 |
| PseudoR$^2$ | 0.0384 | 0.0432 | 0.0540 | 0.0390 | 0.0398 | 0.0638 |

Notes: All t-statistics are presented in parentheses under the estimated coefficient. ***, **, and * indicate 1%, 5% and 10% of significance levels, respectively.

makes it difficult for enterprises to meet the capital demand of innovation projects only by internal financing, and also makes enterprises face high external financing costs. However, the characteristics of innovative project financing of high-tech enterprises are contrary to the principle of "liquidity, security and profitability" adhered to by traditional financial institutions, so the uncertainty of credit availability of high-tech enterprises is higher, and the demand for the new financial model of digital inclusive finance is stronger.

## 7. Mechanism test

### 7.1 Mechanism of information asymmetry reduction

In order to verify the mechanism of information asymmetry reduction of digital finance affecting corporate green innovation, with reference to Niu et al. (2023) [62] and combined with the design of benchmark regression model, this paper constructs the following intermediary effect

models (6), (7) and (8):

$$ASY_{it} = \alpha + \beta_1 DIF_{it} + \sum \varphi CV_{it} + \varepsilon_{it} \tag{6}$$

$$Y_{it} = \alpha + \beta_1 ASY_{it} + \sum \varphi CV_{it} + \varepsilon_{it} \tag{7}$$

$$Y_{it} = \alpha + \beta_1 DIF_{it} + \beta_2 ASY_{it} + \sum \varphi CV_{it} + \varepsilon_{it} \tag{8}$$

The specific regression results are shown in Table 9. The coefficient of *DIF* in Column (1) is significantly positive, indicating that the development of digital finance is conducive to promoting green innovation of enterprises. The coefficient of *DIF* in Column (2) is significantly negative, indicating that the development of digital finance can reduce information asymmetry; The *ASY* coefficients in Columns (3) and (4) are significantly negative at the 1% level, and the results of stepwise regression method in Column (4) show that the *DIF* coefficient is lower than that in Column (1). On this basis, the Sobel test is further conducted in this paper, and it can be found that the Z-value statistic is 13.350, which is significant at 1% level. At the same time, Bootstrap (1000 times) sampling test was conducted in this paper, and it was found that the confidence interval of mediating effect with 95% confidence was [0.0087352, 0.0163977], without 0. The above results indicated that the reduction of information asymmetry played a mediating effect. That is, the development of digital finance will reduce information asymmetry and promote green innovation of enterprises. Hypothesis 2a in this paper is verified.

## 7.2 Mechanism of consumer demand stimulus

In order to verify the consumption structure optimization mechanism of digital finance affecting corporate green innovation, this paper constructs the following intermediary effect models (9), (10) and (11) with reference to the design ideas of intermediary effect mentioned above:

$$struct_{it} = \alpha + \beta_1 DIF_{it} + \sum \varphi CV_{it} + \varepsilon_{it} \tag{9}$$

$$Y_{it} = \alpha + \beta_1 struct_{it} + \sum \varphi CV_{it} + \varepsilon_{it} \tag{10}$$

$$Y_{it} = \alpha + \beta_1 DIF_{it} + \beta_2 struct_{it} + \sum \varphi CV_{it} + \varepsilon_{it} \tag{11}$$

The specific regression results are shown in Table 10. The coefficient of *DIF* in Column (1) is significantly positive, indicating that the development of digital finance is conducive to promoting green innovation of enterprises. The coefficient of *DIF* in Column (2) is significantly positive, indicating that the development of digital finance can optimize the consumption structure; *struct* coefficients in Columns (3) and (4) are significantly positive at the 1% level, and the results of stepwise regression method in Column (4) show that *DIF* coefficient has decreased compared with Column (1). On this basis, the Sobel test is further conducted in this paper, and it can be found that the Z-value statistic is 7.004, which is significant at 1% level. At the same time, Bootstrap (1000 times) sampling test was conducted in this paper, and it was found that the confidence interval of the intermediary effect with 95% confidence was [0.0062394,0.0125897], excluding 0. The above results indicated that the optimization of consumption structure played a mediating effect. That is, the development of digital finance will optimize the consumption structure, thus promoting green innovation of enterprises.

In order to verify the total consumption increased mechanism of digital finance affecting corporate green innovation, this paper constructs the following intermediary effect models

**Table 9. Test results of mediating mechanism: Information asymmetry reduction.**

| | (1) | (2) | (3) | (4) |
|---|---|---|---|---|
| | *PAT* | *ASY* | *PAT* | *PAT* |
| DIF | 0.0313*** | -0.0009*** | | 0.0187*** |
| | (7.0883) | (-14.1697) | | (4.1634) |
| ASY | | | -14.1724*** | -13.9329*** |
| | | | (-7.8536) | (-7.6638) |
| Size | 8.1154*** | -0.2564*** | 4.7640*** | 4.5431*** |
| | (12.5762) | (-31.5685) | (10.7245) | (10.4924) |
| Age | -13.6774*** | 0.0527*** | -11.1946*** | -12.9431*** |
| | (-7.6385) | (4.7335) | (-7.2501) | (-7.3819) |
| lev | 0.2798 | 0.4086*** | 5.9037*** | 5.9733*** |
| | (0.1584) | (15.9915) | (4.2492) | (4.2677) |
| Top1 | 12.2071** | 0.5130*** | 19.3474*** | 19.3542*** |
| | (3.1316) | (16.1489) | (4.8995) | (4.9027) |
| Mfee | 0.0037*** | 0.0001 | 0.0035** | 0.0044*** |
| | (3.3422) | (0.8571) | (2.8453) | (3.7408) |
| Growth | 0.4388 | -0.0002 | 0.4361 | 0.4366 |
| | (1.8886) | (-0.0132) | (1.6771) | (1.7417) |
| ROE | 1.5939* | -0.1869*** | -1.5422 | -1.0097 |
| | (2.1068) | (-4.0271) | (-1.7856) | (-1.2131) |
| TobinQ | -1.7457*** | -0.1341*** | -3.2494*** | -3.6145*** |
| | (-4.6938) | (-11.2540) | (-7.1932) | (-7.5258) |
| Mshare | -22.9440*** | 0.1560 | -19.4900*** | -20.7704*** |
| | (-5.8281) | (1.7574) | (-4.8799) | (-5.0984) |
| Indep | 40.0440*** | -0.5546*** | 31.7705*** | 32.3172*** |
| | (4.1228) | (-6.1272) | (3.3574) | (3.4012) |
| Dual | 2.5448** | -0.0207* | 2.3310** | 2.2564** |
| | (3.1065) | (-2.4813) | (3.0127) | (2.8891) |
| SOE | 1.2175* | 0.0467*** | 1.4892* | 1.8678** |
| | (2.0381) | (5.5208) | (2.3985) | (2.9909) |
| Board | -0.4595 | -0.0327 | -1.7738 | -0.9156 |
| | (-0.1909) | (-1.1861) | (-0.7477) | (-0.3888) |
| Constant | -1.6e+02*** | 5.6298*** | -81.4697*** | -77.5154*** |
| | (-11.9723) | (36.6899) | (-7.8876) | (-7.5698) |
| Sobel Z | | 13.350*** | | |
| Bootstrap lower | | 0.0087352 | | |
| Bootstrap upper | | 0.0163977 | | |
| N | 22781 | 22781 | 22781 | 22781 |
| F | 23.2996 | 243.7511 | 24.5772 | 23.2654 |
| Adj_R$^2$ | 0.0696 | 0.3344 | 0.0907 | 0.0912 |

Notes: All t-statistics are presented in parentheses under the estimated coefficient. ***, **, and * indicate 1%, 5% and 10% of significance levels, respectively.

(12), (13) and (14) according to the design ideas of intermediary effect mentioned above:

$$total_{it} = \alpha + \beta_1 DIF_{it} + \sum \varphi CV_{it} + \varepsilon_{it} \tag{12}$$

$$Y_{it} = \alpha + \beta_1 total_{it} + \sum \varphi CV_{it} + \varepsilon_{it} \tag{13}$$

**Table 10. Test results of mediating mechanism: Consumption structure optimization.**

| | (1) | (2) | (3) | (4) |
|---|---|---|---|---|
| | *PAT* | *struct* | *PAT* | *PAT* |
| DIF | 0.0186*** | 135.4656*** | | 0.0092*** |
| | (11.1305) | (110.9647) | | (4.3799) |
| struct | | | 0.0001*** | 0.0001*** |
| | | | (9.9015) | (5.7971) |
| Size | 4.1842*** | 560.0970*** | 4.2197*** | 4.1453*** |
| | (20.9486) | (7.4748) | (21.6741) | (20.8302) |
| Age | -5.2247*** | -3.9e+03*** | -4.3106*** | -4.9522*** |
| | (-11.9460) | (-17.8263) | (-10.3081) | (-11.7184) |
| lev | 1.4406** | -1.4e+03*** | 1.5204** | 1.5379** |
| | (2.9716) | (-4.3928) | (3.1658) | (3.1830) |
| Top1 | 1.4172 | -3.5e+03*** | 1.7045 | 1.6583 |
| | (1.2939) | (-6.5394) | (1.5541) | (1.5094) |
| Mfee | 0.0008 | -2.0399** | 0.0007 | 0.0010* |
| | (1.8704) | (-3.0806) | (1.6299) | (2.2057) |
| Growth | 0.1270 | 1.0e+03*** | 0.0292 | 0.0552 |
| | (1.1601) | (5.2728) | (0.2803) | (0.5231) |
| ROE | 1.4586*** | 877.8716 | 1.2230*** | 1.3976*** |
| | (3.8778) | (1.2402) | (3.5045) | (3.8720) |
| TobinQ | -1.3231*** | -1.4e+02 | -1.1858*** | -1.3136*** |
| | (-8.2853) | (-1.2065) | (-7.9283) | (-8.1830) |
| Mshare | -8.4127*** | 2.7e+03 | -8.2830*** | -8.6035*** |
| | (-4.4977) | (1.7230) | (-4.3973) | (-4.5745) |
| Indep | 14.9400*** | -6.6e+02 | 14.8730*** | 14.9860*** |
| | (5.2012) | (-0.4338) | (5.1822) | (5.2246) |
| Dual | 1.2389*** | 1.1e+03*** | 1.1614*** | 1.1636*** |
| | (4.4963) | (6.3788) | (4.1865) | (4.1957) |
| SOE | -0.0364 | 484.1998** | -0.2042 | -0.0701 |
| | (-0.1480) | (2.8515) | (-0.8338) | (-0.2856) |
| Board | 1.0380 | -1.1e+03* | 0.8754 | 1.1142 |
| | (1.2378) | (-2.4838) | (1.0425) | (1.3261) |
| Constant | -84.9128*** | 4.7e+03* | -87.0219*** | -85.2428*** |
| | (-19.9255) | (2.4621) | (-20.9770) | (-19.9393) |
| Sobel Z | | 7.004*** | | |
| Bootstrap lower | | 0.0062394 | | |
| Bootstrap upper | | 0.0125897 | | |
| N | 22781 | 22781 | 22781 | 22781 |
| F | 70.7043 | 1.3e+03 | 62.2080 | 68.7575 |
| Adj_R² | 0.1367 | 0.4638 | 0.1379 | 0.1385 |

Notes: All t-statistics are presented in parentheses under the estimated coefficient. ***, **, and * indicate 1%, 5% and 10% of significance levels, respectively.

$$Y_{it} = \alpha + \beta_1 DIF_{it} + \beta_2 total_{it} + \sum \varphi CV_{it} + \varepsilon_{it} \qquad (14)$$

The specific regression results are shown in Table 11. The coefficient of *DIF* in Column (1) is significantly positive, indicating that the development of digital finance is conducive to

**Table 11. Test results of mediating mechanism: Total consumption increased.**

|  | (1) | (2) | (3) | (4) |
|---|---|---|---|---|
|  | *PAT* | *total* | *PAT* | *PAT* |
| DIF | 0.0186*** | 81.0888*** |  | 0.0085*** |
|  | (11.1305) | (110.1486) |  | (4.0808) |
| total |  |  | 0.0002*** | 0.0001*** |
|  |  |  | (10.6382) | (6.6021) |
| Size | 4.1842*** | 322.9818*** | 4.2158*** | 4.1439*** |
|  | (20.9486) | (6.9499) | (21.6880) | (20.8193) |
| Age | -5.2247*** | -2.4e+03*** | -4.3188*** | -4.9233*** |
|  | (-11.9460) | (-17.2275) | (-10.3692) | (-11.6535) |
| lev | 1.4406** | -9.0e+02*** | 1.5358** | 1.5524** |
|  | (2.9716) | (-4.5432) | (3.2039) | (3.2190) |
| Top1 | 1.4172 | -2.0e+03*** | 1.7091 | 1.6722 |
|  | (1.2939) | (-6.3628) | (1.5607) | (1.5245) |
| Mfee | 0.0008 | -1.6385*** | 0.0008 | 0.0011* |
|  | (1.8704) | (-3.6517) | (1.8006) | (2.3266) |
| Growth | 0.1270 | 713.5912*** | 0.0116 | 0.0380 |
|  | (1.1601) | (5.3409) | (0.1111) | (0.3615) |
| ROE | 1.4586*** | 590.7497 | 1.2184*** | 1.3850*** |
|  | (3.8778) | (1.2399) | (3.4868) | (3.8473) |
| TobinQ | -1.3231*** | -1.2e+02* | -1.1854*** | -1.3079*** |
|  | (-8.2853) | (-1.9892) | (-7.9907) | (-8.1810) |
| Mshare | -8.4127*** | 1.1e+03 | -8.2149*** | -8.5449*** |
|  | (-4.4977) | (1.1642) | (-4.3683) | (-4.5504) |
| Indep | 14.9400*** | -69.4059 | 14.8283*** | 14.9486*** |
|  | (5.2012) | (-0.0734) | (5.1690) | (5.2144) |
| Dual | 1.2389*** | 804.8435*** | 1.1325*** | 1.1386*** |
|  | (4.4963) | (7.6436) | (4.0855) | (4.1085) |
| SOE | -0.0364 | 291.1055** | -0.1992 | -0.0727 |
|  | (-0.1480) | (2.7475) | (-0.8138) | (-0.2963) |
| Board | 1.0380 | -7.1e+02** | 0.9010 | 1.1271 |
|  | (1.2378) | (-2.5923) | (1.0732) | (1.3416) |
| Constant | -84.9128*** | 6.2e+03*** | -87.4946*** | -85.6904*** |
|  | (-19.9255) | (5.1605) | (-21.0872) | (-19.9690) |
| Sobel Z |  | 7.798*** |  |  |
| Bootstrap lower |  | 0.0071948 |  |  |
| Bootstrap upper |  | 0.0130326 |  |  |
| N | 22781 | 22781 | 22781 | 22781 |
| F | 70.7043 | 1.3e+03 | 62.3932 | 68.0055 |
| Adj_R$^2$ | 0.1367 | 0.4452 | 0.1384 | 0.1390 |

Notes: All t-statistics are presented in parentheses under the estimated coefficient. ***, **, and * indicate 1%, 5% and 10% of significance levels, respectively.

promoting green innovation of enterprises. The coefficient of *DIF* in Column (2) is significantly positive, indicating that the development of digital finance can increase the total consumption; The *total* coefficients in Columns (3) and (4) are significantly positive at the 1% level, and the results of stepwise regression method in Column (4) show that the *DIF* coefficient has decreased compared with that in Column (1). On this basis, the Sobel test is further

conducted in this paper, and it can be found that the Z-value statistic is 7.798, which is significant at 1% level. At the same time, Bootstrap (1000 times) sampling test was conducted in this paper, and it was found that the confidence interval of the mediating effect with 95% confidence was [0.0071948, 0.0130326], excluding 0. The above results indicated that the increase in total consumption played a mediating effect. That is, the development of digital finance will increase the total consumption and thus promote the green innovation of enterprises. Hypothesis 2b in this paper is verified.

## 7.3 Mechanism of factor market distortion mitigation

In order to verify the mitigation mechanism of factor market distortion of digital finance affecting enterprise green innovation, this paper constructs the following intermediary effect models (15), (16) and (17) according to the design ideas of intermediary effect mentioned above:

$$Dist_{it} = \alpha + \beta_1 DIF_{it} + \sum \varphi CV_{it} + \varepsilon_{it} \tag{15}$$

$$Y_{it} = \alpha + \beta_1 Dist_{it} + \sum \varphi CV_{it} + \varepsilon_{it} \tag{16}$$

$$Y_{it} = \alpha + \beta_1 DIF_{it} + \beta_2 Dist_{it} + \sum \varphi CV_{it} + \varepsilon_{it} \tag{17}$$

The specific regression results are shown in Table 12. The coefficient of *DIF* in Column (1) is significantly positive, indicating that the development of digital finance is conducive to promoting green innovation of enterprises. The coefficient of *DIF* in Column (2) is significantly negative, indicating that the development of digital finance can alleviate factor market distortion; *Dist* coefficients in Columns (3) and (4) are significantly negative at the 1% level, and the results of stepwise regression method in Column (4) show that *DIF* coefficient is lower than that in Column (1). On this basis, the Sobel test is further conducted in this paper, and it can be found that the Z-value statistic is 3.608, which is significant at 1% level. At the same time, Bootstrap (1000 times) sampling test was conducted in this paper, and it was found that the confidence interval of mediating effect with 95% confidence was [0.0005324, 0.0017878], excluding 0. The above results indicated that mitigation of factor market distortion played a mediating effect. In other words, the development of digital finance will alleviate the distortion of factor market, thus promoting the green innovation of enterprises. The hypothesis 2c in this paper is verified.

## 7.4 The impact of intellectual property protection

This study further evaluated the effect of intellectual property protection (*IPP*) on the relationship between digital finance and enterprise green innovation by adding the interaction term of digital finance and intellectual property protection (*DIF×IPP*), and the regression results are shown in Panel A of Table 13. The regression coefficients of the interaction term *DIF×IPP* on the total green innovation level and green utility model innovation of enterprises are all significantly positive at the 1% level, the regression coefficients of the interaction term *DIF×IPP* on the green invention innovation of enterprises are all significantly positive at the 5% level, and the regression results indicate that the higher the protection of intellectual property rights, the stronger the promotion effect of digital finance on enterprise green innovation, enterprise green invention innovation, and green utility model innovation. This indicates that for a firm with the intention of green innovation, strong intellectual property protection can buffer the potential loss caused by the copying or stealing of the firm's green patents, which in turn enhances the firm's confidence in sustainable green innovation with the help of digital finance. Hypothesis 3a is verified.

**Table 12. Test results of mediating mechanism: Factor market distortion mitigation.**

| | (1) | (2) | (3) | (4) |
|---|---|---|---|---|
| | *PAT* | *Dist* | *PAT* | *PAT* |
| DIF | 0.0313*** | -0.0117*** | | 0.0301*** |
| | (7.0883) | (-8.4393) | | (7.0359) |
| Dist | | | -0.1074*** | -0.0988*** |
| | | | (-4.0957) | (-3.8467) |
| Size | 8.1154*** | -0.1563* | 8.5569*** | 8.1000*** |
| | (12.5762) | (-1.9902) | (13.0801) | (12.5693) |
| Age | -13.6774*** | -1.7861*** | -11.0515*** | -13.8538*** |
| | (-7.6385) | (-6.1383) | (-6.8627) | (-7.5919) |
| lev | 0.2798 | 1.8678*** | 0.2084 | 0.4643 |
| | (0.1584) | (4.6870) | (0.1185) | (0.2614) |
| Top1 | 12.2071** | 2.5060*** | 12.2657** | 12.4547** |
| | (3.1316) | (4.1302) | (3.1363) | (3.1774) |
| Mfee | 0.0037*** | -0.0037*** | 0.0018 | 0.0033** |
| | (3.3422) | (-5.0224) | (1.6904) | (3.1670) |
| Growth | 0.4388 | -0.9792*** | 0.3329 | 0.3421 |
| | (1.8886) | (-6.4549) | (1.5144) | (1.5502) |
| ROE | 1.5939* | 0.6982 | 0.8772 | 1.6629* |
| | (2.1068) | (1.3792) | (1.2017) | (2.2572) |
| TobinQ | -1.7457*** | -0.1407 | -1.1159*** | -1.7596*** |
| | (-4.6938) | (-1.3637) | (-3.4654) | (-4.7340) |
| Mshare | -22.9440*** | -3.7953* | -21.3338*** | -23.3189*** |
| | (-5.8281) | (-2.4277) | (-5.5405) | (-5.8691) |
| Indep | 40.0440*** | 0.9431 | 39.4736*** | 40.1372*** |
| | (4.1228) | (0.5288) | (4.0709) | (4.1306) |
| Dual | 2.5448** | 1.0256*** | 2.7840*** | 2.6461** |
| | (3.1065) | (4.8702) | (3.4510) | (3.2638) |
| SOE | 1.2175* | -2.9981*** | 0.2628 | 0.9213 |
| | (2.0381) | (-15.8264) | (0.4668) | (1.5801) |
| Board | -0.4595 | 1.4340** | -1.6860 | -0.3179 |
| | (-0.1909) | (2.7759) | (-0.6898) | (-0.1316) |
| Constant | -1.6e+02*** | 21.5859*** | -1.6e+02*** | -1.5e+02*** |
| | (-11.9723) | (10.3468) | (-12.5865) | (-11.9501) |
| Sobel Z | | 3.608*** | | |
| Bootstrap lower | | 0.0005324 | | |
| Bootstrap upper | | 0.0017878 | | |
| N | 22781 | 22781 | 22781 | 22781 |
| F | 23.2996 | 62.4946 | 24.4411 | 23.8649 |
| Adj_R2 | 0.0696 | 0.0352 | 0.0688 | 0.0702 |

Notes: All t-statistics are presented in parentheses under the estimated coefficient. ***, **, and * indicate 1%, 5% and 10% of significance levels, respectively.

## 7.5 The impact of environmental governance

This study further evaluated the effect of environmental governance (*FREQ*) on the relationship between digital finance and enterprise green innovation by adding the interaction term of digital finance and environmental governance (*DIF×FREQ*), and the regression results are shown in Panel B of Table 13. The regression coefficients of the interaction term *DIF×FREQ*

**Table 13. Impact of intellectual property protection and environmental governance.**

|  | (1) | (2) | (3) |
|---|---|---|---|
|  | *PAT* | *INPAT* | *UPAT* |
| Panel A: Impact of intellectual property protection. | | | |
| *DIF×IPP* | 7.4315*** | 3.9876** | 3.4439*** |
|  | (4.0553) | (3.2631) | (4.4496) |
| *DIF* | -7.3888*** | -3.9614** | -3.4274*** |
|  | (-4.0411) | (-3.2489) | (-4.4385) |
| *IPP* | -1200** | -520 | -700*** |
|  | (-2.8653) | (-1.7880) | (-4.0607) |
| *Control* | Yes | Yes | Yes |
| N | 22781 | 22781 | 22781 |
| F | 21.5266 | 18.4670 | 24.7870 |
| Adj_R² | 0.0702 | 0.0526 | 0.0776 |
| Panel B: Impact of environmental governance. | | | |
| *DIF×FREQ* | 15.4763*** | 8.1936** | 7.2827*** |
|  | (3.8678) | (2.9655) | (4.6061) |
| *DIF* | -0.0163 | -0.0059 | -0.0104** |
|  | (-1.5358) | (-0.7743) | (-2.6758) |
| *FREQ* | -1800** | -830* | -980 *** |
|  | (-3.0195) | (-2.0343) | (-3.9916) |
| *Control* | Yes | Yes | Yes |
| N | 22781 | 22781 | 22781 |
| F | 21.3677 | 18.3372 | 24.5045 |
| Adj_R² | 0.0719 | 0.0536 | 0.0797 |

Notes: All t-statistics are presented in parentheses under the estimated coefficient. ***, **, and * indicate 1%, 5% and 10% of significance levels, respectively.

on the total green innovation level, the green invention innovation and green utility model innovation of enterprises are all significantly positive at the 1% level, and the regression coefficient of the interaction term *DIF×FREQ* on enterprise green utility innovation is significantly positive at the 1% level, which indicates that the higher the environmental governance of the region where the enterprise is located, the higher the promotion of digital finance on enterprise green innovation, enterprise green invention innovation, and green utility innovation. Environmental governance increases the production and operation costs of enterprises, forcing them to conduct green innovation and achieve green development and transformation and upgrading of industries. Hypothesis 3b is verified.

# 8. Conclusions and policy implications

## 8.1 Conclusions

In this study, the theoretical mechanism of how digital finance influences enterprise green innovation was systematically analyzed. We used panel data from 2071 A-share listed companies in China from 2011 to 2021 to empirically evaluate the impact of digital finance and its coverage breadth, usage depth, and digitization level on enterprise green innovation. Meanwhile, the heterogeneous effects of digital finance on green innovation of enterprises with different characteristics are investigated. Furthermore, the mechanisms of information asymmetry reduction, consumer demand stimulus, and factor market distortion mitigation by which digital finance affects enterprise green innovation and the impact of intellectual

property protection and environmental governance on the relationship between digital finance and enterprise green innovation were further investigated. The main conclusions of this paper are as follows:

First, the development of digital finance has a significant role in promoting the green innovation of enterprises, which is specifically manifested in encouraging the green invention innovation and green utility model innovation of enterprises. Compared with green utility model innovation, digital finance has a stronger incentive effect on green invention and innovation. The coverage breadth of digital finance has a significant positive impact on enterprise green innovation, enterprise green invention innovation and enterprise green utility model innovation. The usage depth of digital finance only has a significant impact on enterprises' green invention and innovation; The digitalization level of digital finance has no significant impact on enterprises' green innovation. These results are different from those of some scholars, such as Li et al. (2023) [25]. They believe that the promotion of green innovation by the development of digital inclusive finance is mainly driven by the depth of application of digital inclusive finance and the digitalization of inclusive finance. These results are also similar to the studies of some scholars, such as Fan et al. (2022) found that the coverage breadth and usage depth of digital finance can promote the green innovation of enterprises, and the degree of digitalization has no significant impact on the green technology innovation of enterprises, but it lacks the fractal dimension of digital finance.

Second, the impact of digital finance on promoting green innovation of enterprises is heterogeneous. The development of digital finance and its coverage breadth and usage depth only has a significant impact on the green innovation, green invention innovation and green utility model innovation of small and medium-sized enterprises and high-tech enterprises. In addition, the digitalization level of digital finance only has a significant promoting effect on the green innovation and green invention innovation of small and medium-sized enterprises and high-tech enterprises. The existing literature lacks the heterogeneity research on green innovation of enterprises with different scale of influence of digital finance and whether they have high-tech qualifications.

Third, digital finance promotes corporate green innovation by reducing information asymmetry, stimulating consumer demand, and alleviating regional factor market distortions. By strengthening intellectual property protection and environmental governance, the role of digital finance in promoting green innovation can be further strengthened.

## 8.2 Policy implications

According to this research, we promote the following policy implications. From the national perspective, China and other emerging economies should increase their investment in digital infrastructure and, with the help of big data, artificial intelligence, cloud computing and other digital technologies, set up green innovation databases and assessment systems to collect, monitor, calculate and analyse real-time information on the green innovation activities of enterprises, so as to provide financial institutions with the relevant information needed to assess the green innovation capacity and environmental performance of enterprises, so that financial institutions can better support enterprises' green innovations in terms of financing and investment. Meanwhile, China and other emerging economies should promote resource integration and cooperation among enterprises, financial institutions, research institutes and other parties, and set up a collaborative mechanism to jointly research and develop green patented technologies, promote green digital financial products, and carry out demonstration projects, so as to realize optimal allocation of resources and collaborative innovation. Finally, China and other emerging economies should comprehensively promote the marketization process, give full

play to the decisive role of the market mechanism in resource allocation, reduce biased financial policies, provide higher-quality digital financial services for green innovation and development, and guide enterprises and scientific research institutes to invest their capital elements in green innovation and R&D activities.

From the perspective of financial institutions, financial institutions should rely on digital finance to realize the digital function of financial infrastructure, use digital technology to strengthen the intelligent identification capability of green enterprises and green projects, and actively provide enterprises with diversified financial products and services such as green credit, green bonds, etc. At the same time, digital financial platforms should guide residents to form a green consumption system through the intelligent recommendation of green products, personalized service of green consumption assessment report, and provision of green consumption. Meanwhile, digital financial platforms should guide residents to form green consumption concepts and low-carbon lifestyles through the intelligent recommendation of green products, personalized services of green consumption assessment reports, and the provision of green consumption credits, thus encouraging enterprises to accelerate green innovation to meet consumer demand. Finally, Financial institutions should develop differentiated digital financial products and provide personalized digital financial services for enterprises of different sizes and industries.

## 9. Limitations and future research

This paper also has some limitations that warrant further research in the future. First, this study only explores the impact of digital finance on corporate green innovation from the perspectives of reducing information asymmetry, stimulating consumer demand, and mitigating factor market distortions, and future research can explore the realization path of digital finance on corporate green innovation from other perspectives, such as technology spillovers. Second, due to the limitation of data sources, this study only tested the impact of digital finance on corporate green innovation during the period of 2011–2021, and future researchers can extend the timeframe of the study to test the relationship between the two in a longer timeframe. Third, the study samples are all Chinese listed companies, and a large number of Chinese non-listed companies and companies from other countries are not included in the review, and the role of digital finance in promoting green innovation for companies from other countries or Chinese non-listed SMEs needs to be further explored in the future. Finally, the conclusions of this paper come from the quantitative analysis of a large amount of data, and it is difficult to conduct in-depth research on the evolutionary process between variables as in case studies. In the future, researchers can conduct detailed case studies on the green innovation activities of different types of enterprises to reveal the evolutionary process of digital finance affecting the green innovation of enterprises.

## Supporting information

**S1 Dataset.**
(XLSX)

**S1 File.**
(DO)

**S1 Table.**
(DOCX)

## Author Contributions

**Conceptualization:** Linzhi Han, Zhongan Zhang.

**Data curation:** Zhongan Zhang.

**Formal analysis:** Zhongan Zhang.

**Funding acquisition:** Linzhi Han.

**Investigation:** Zhongan Zhang.

**Methodology:** Zhongan Zhang.

**Project administration:** Linzhi Han.

**Resources:** Zhongan Zhang.

**Software:** Zhongan Zhang.

**Supervision:** Linzhi Han.

**Validation:** Zhongan Zhang.

**Visualization:** Zhongan Zhang.

**Writing – original draft:** Zhongan Zhang.

**Writing – review & editing:** Zhongan Zhang.

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
