## [Decision Letter · Decision Letter 0]

10 Jul 2023

PONE-D-23-15681The role of digital finance in driving green innovation: A study of listed companies in ChinaPLOS ONE

Dear Dr. Zhang,

Thank you for submitting your manuscript to PLOS ONE. After careful consideration, we feel that it has merit but does not fully meet PLOS ONE’s publication criteria as it currently stands. Therefore, we invite you to submit a revised version of the manuscript that addresses **all** the points raised during the review process.

You may ignore the comment about novelty. However, feel free to address it if you wish to do so. 

We look forward to receiving your revised manuscript.

Kind regards,

Zakaria Boulanouar, PhD

Academic Editor

PLOS ONE

Journal Requirements:

Additional Editor Comments:

Overall, I find the topic of the manuscript to be both interesting and relevant to our readership. Additionally, the comments provided by the reviewers are encouraging and offer valuable insights for improvement. However, I would like to draw your attention to certain concerns including the methodology employed in the study, the justification for using it and implications and policy recommendations. It is evident that major revisions are necessary to address these issues and allow you to conduct the necessary adjustments. By addressing these concerns, your manuscript has potential to significantly enhance its scientific rigor and validity.

Reviewers' comments:

Reviewer's Responses to Questions

**Comments to the Author**

1. Is the manuscript technically sound, and do the data support the conclusions?

Reviewer #1: Yes

Reviewer #2: Yes

2. Has the statistical analysis been performed appropriately and rigorously? 

Reviewer #1: No

Reviewer #2: Yes

3. Have the authors made all data underlying the findings in their manuscript fully available?

Reviewer #1: Yes

Reviewer #2: Yes

4. Is the manuscript presented in an intelligible fashion and written in standard English?

Reviewer #1: Yes

Reviewer #2: Yes

5. Review Comments to the Author

Reviewer #1: I would appreciate the editor for providing me an opportunity to review the article “The role of digital finance in driving green innovation: A study of listed companies in China”. In this paper, the authors examined the effect of digital finance in promoting green innovation based on a dataset collected from listed firms in China. The paper is overall interesting and well-organized. However, I am a bit concerned of the novelty and the experimental design of this paper.

1. The relationship between digital finance and green innovation has been explored in some earlier studies, such as https://doi.org/10.1007/s11356-022-18667-4, https://doi.org/10.1007/s11356-022-19785-9, and https://doi.org/10.1007/s11356-022-21802-w. I would strongly recommend the authors to add a separate paragraph to clarify what makes this paper different from the previous ones.

2. The literature review of this paper is too short at the present form. I think the authors should extend it.

3. The authors should explain what does the star symbols mean in Tables. Moreover, are the figures in paratheses the t-statistics? The authors should not let the readers guess the contents in the tables.

4. The model specification for Section 7 is unclear and I doubt the stepwise mediation regression models are mis-specified in this paper.

Reviewer #2: In this paper, the authors use econometric tools to investigate the effects of digital finance on green innovation and deepen the research on the incentive effects of digital finance on corporate green innovation and impact mechanisms using micro-level control variables and macro-level control variables for robustness test. Here are some suggestions:

1. Title should be redesigned. The role of digital finance in driving green innovation… is very huge, it is not precisely summarized what you have done.

2. Why the analysis is limited by 2020 year? I suppose that 2021 data is already available in mentioned sources.

3. - Research gaps and contribution should be well mentioned in the introduction. A good research gap can give the reader more insight

4. The descriptive statistics should check for normality and for the asymmetry of the distributions of the variables. Distortions can be calculated using skewness and kurtosis( Min=median=0 for many variables).

5. The interpretation of the tables3, 4 is superficial. You only mention the results sign and significance and the validation of the hypothesis. What does it mean for the chineese A-share listed firms ?what are the implications for Chinese government policy? Does it have high standard implication for Chinese economy?

6. Useless tests of robustness via independent variable lag.(page 10). For example you can select the lag using schwartz or Akaike information criterion.

7. The heterogeneity analysis results discussions aren’t based on previous studies. Why SMEs green innovations are more impacted by the digital finance while that impact isn’t significant for large enterprises?

8. How you compare and contrast your research findings with other studies?

9. Section VIII lacks enough policy implications. The policy suggestions are also lack of enough practicability. Authors should put forward detailed policy suggestions which apply to China and to many countries, especially emerging countries.

10. The possible future research directions should be added at the end of the paper, so that in the future the researchers who are interested in this important topic could carry out follow-up studies.

Minor

10. Text requires polishing

11.Typing repetition p6 (environment protection)

6. PLOS authors have the option to publish the peer review history of their article (what does this mean?). If published, this will include your full peer review and any attached files.

Reviewer #1: No

Reviewer #2: No

---

## [Author Response · Author response to Decision Letter 0]

14 Aug 2023

Responses to reviewers

(Original comments by the academic editor and reviewers are in blue color, the page and line numbers of modify content in the “Revised Manuscript with Track Changes” are marked in red color)

Additional Editor Comments: Overall, I find the topic of the manuscript to be both interesting and relevant to our readership. Additionally, the comments provided by the reviewers are encouraging and offer valuable insights for improvement. However, I would like to draw your attention to certain concerns including the methodology employed in the study, the justification for using it and implications and policy recommendations. It is evident that major revisions are necessary to address these issues and allow you to conduct the necessary adjustments. By addressing these concerns, your manuscript has potential to significantly enhance its scientific rigor and validity.

Reply:

We are very grateful to you for pointing out this valuable and very central issues, which is important for refining and improving the quality of the whole article. We explain in detail the methodology and the justification for using it (Line 10, page 10 - Line 9, page 14), and have made more specific implications and policy recommendations (Line 4, page 33 - Line 22, page 35).

Reviewer #1:

1. Comment: The relationship between digital finance and green innovation has been explored in some earlier studies, such as https://doi.org/10.1007/s11356-022-18667-4, https://doi.org/10.1007/s11356-022-19785-9, and https://doi.org/10.1007/s11356-022-21802-w. I would strongly recommend the authors to add a separate paragraph to clarify what makes this paper different from the previous ones.

1. Reply:

We are very grateful to the reviewers for pointing this out. We have added a separate paragraph to clarify what makes this paper different from the previous ones in the introduction (Line 18, page 2 - Line 41, page 2). Modification details are as follows:

1). the main contribution in the introduction

The main contributions of this study to the existing literature are summarized as follows: First, most existing related literatures have explored the mechanism of digital finance on enterprise green innovation from the perspectives of alleviating financing constraint(Liu et al., 2022; Fan et al., 2022; Liu et al., 2022; Kong et al., 2022; Xue et al., 2022; Li et al., 2022; Li et al., 2023; Ma et al., 2023), improving corporate transparency(Rao et al.,2022), increasing R&D investment(Liu et al., 2022; Li et al., 2023), enhancing financial flexibility(Fan et al., 2022), improving environmental information disclosure quality(Kong et al., 2022), mitigating financial mismatch(Li et al., 2023), improving internal control(Liu et al., 2022; Ma et al., 2023), and solving internal and external information constraint(Liu et al., 2022), whereas our study theoretically analyzes and empirically examines the mechanisms of information asymmetry re-duction, consumer demand stimulus and factor market distortion mitigation by which digital finance affects enterprise green innovation, as well as the impacts of intellectual property protection and environmental governance on the relationship between digital finance and enterprise green innovation, which sheds new light on digital finance influencing enterprise green innovation. Second, most existing studies on the heterogeneity of the impact of digital finance on enterprise green innovation are based on the level of regional economic development(Liu et al., 2022; Li et al., 2022; Rao et al., 2022), the degree of pollution in the industry(Liu et al., 2022; Li et al., 2022), the nature of enterprise property rights(Rao et al., 2022; Kong et al., 2022; Li et al., 2022; Li et al., 2023), the life cycle of the enterprise(Fan et al., 2022; Rao et al., 2022; Xue et al., 2022), the level of regional financial development(Li et al., 2023) and the intensity of regional financial regulation(Li et al., 2023), whereas our study analyzes the heterogeneity of the results of the study from the perspectives of enterprise scale and whether they possess high-tech qualifications, and enriches the study on the impact of digital finance on the green innovation of enterprises with different characteristics.

2. Comment: The literature review of this paper is too short at the present form. I think the authors should extend it.

2. Reply:

We are very grateful to you for pointing out this valuable and very central issue, which is important for refining and improving the quality of the whole article. We expanded and revised the literature review of the paper and added the latest literature related to digital finance and corporate green innovation (Line 48, page 2 - Line 2, page 4). Modification details are as follows:

1). Literature Review

In recent years, more and more scholars pay attention to the impact of digital finance on green innovation. Liu et al. (2022) [18] found that the development of digital finance, its coverage breadth and usage depth will increase the number of green patents granted by enterprises, especially the number of green invention patents granted, and significantly increase the quantity and quality of green innovations, and this effect is more obvious in the eco-nomically backward areas and highly polluted industries. Fan et al. (2022) [19] found that digital finance and its coverage breadth and usage depth effectively promote corporate green innovation, but the degree of digital finance digitization has no significant effect on corporate green technology innovation, and digital finance only has a significant positive impact on the green technology innovation of enterprises with high financing constraints and high financial leverage groups, industries with low concentration, growth, and low quality of environmental disclosure reports and environmental governance reports. Liu et al. (2022) [20] found that digital finance can stimulate enterprises' green innovation by increasing the coverage breadth and usage depth of digital finance, and the impact is stronger when the analyst optimism bias is lower and the synchronization is higher. Kong et al. (2022) [21] found that digital financial institutions can alleviate the information asymmetry in the green innovation market and directly promote the green innovation behavior of enterprises through digital technologies such as big data analysis of enterprise behavior, and digital finance has a more prominent role in the promotion of green innovation in large state-owned enterprises. Xue et al. (2022) [22] found that digital finance can promote the green innovation of enterprises in the heavy pollution industry, and the impact on the green innovation of heavy pollution enterprises in the maturity period is higher than that of enterprises in the growth period. Li et al. (2022) [23] found that the promotion effect of digital finance on enterprise green innovation persists and shows an upward trend over time, and with the increasing level of digital finance development, its impact on green innovation is more significant, and this effect is more obvious in state-owned enterprises, economically developed regions in the east, and high-pollution industries. Rao et al. (2022) [24] found that digital finance can significantly increase the number of corporate green patent applications, the number of citations of green patents, and improve the quantity and quality of corporate green innovations, and this effect is stronger in eastern, state-owned, and mature firms. Li et al. (2023) [25] found that the promotion of digital finance on green innovation is mainly driven by the developmental drivers of the depth of use and level of digitization of digital finance. Li et al. (2023) [26] found that the effect of digital finance on green innovation is more obvious in state-owned enterprises and in regions with a lower degree of financial development and stronger financial regulation. Ma et al. (2023) [27] found that digital finance and its coverage breadth, usage depth and digitization level can significantly improve the level of green innovation of enterprises. 

Secondly, some literatures have investigated the mechanism of digital finance on corporate green innovation, which mainly includes the mechanism of financing constraint alleviation, R&D investment increase, financial flexibility enhancement, environmental information disclosure quality improvement, financial mismatch mitigation, internal control improvement and internal and external information constraint dissolution, etc. Liu et al. (2022) [18] and Li et al. (2023) [26] found that digital finance promotes green innovation by alleviating capital constraints and increasing R&D investment. Fan et al. (2022) [19] found that digital finance can improve internal financing for corporate green technology innovation by reducing financing costs and increasing financial flexibility. Liu et al. (2022) [20] found that digital finance can reduce the financial constraints of enterprises by providing them with loans and improving cash flow, improve transparency by addressing external information constraints, and increase internal control and research investment, thus making enterprises more and more willing to carry out green innovation. Kong et al. (2022) [21] found that digital finance indirectly promotes green innovation by improving the quality of enterprises' environmental information disclosure and reducing financial constraints. Xue et al. (2022) [22] found that digital finance promotes green innovation by alleviating corporate financing constraints and financial mismatches. Li et al. (2022) [23] found that digital finance can improve green innovation by reducing corporate financing constraints and improving the overall innovation capacity of cities. Rao et al. (2022) [24] found that the development of digital finance can promote corporate green innovation by improving the transparency of enter-prises, increasing the efficiency of inter-enterprise capital flows, and making the allocation of financial resources more convenient. Li et al. (2023) [25] found that digital finance can promote corporate green innovation by im-proving the efficiency of financial services and alleviating capital misallocation. Ma et al.(2023) [27] found that digital finance can improve the level of green technology innovation of enterprises by easing financing constraints and improving internal control.

In summary, the existing literature has studied the relationship between digital finance and corporate green innovation from different perspectives, and has achieved many valuable results, but there is still much room for expansion. First, the existing relevant literature has only explored the role mechanism of digital finance on corporate green innovation from the perspectives of alleviating financing constraints(Liu et al., 2022 [18]; Fan et al., 2022 [19]; Liu et al., 2022 [20]; Kong et al., 2022 [21]; Xue et al., 2022 [22]; Li et al., 2022 [23]; Li et al., 2023 [26]; Ma et al., 2023 [27]), improving corporate transparency(Rao et al., 2022 [24]), increasing R&D investment(Liu et al., 2022 [18]; Li et al., 2023 [26]), enhancing financial flexibility(Fan et al., 2022 [19]), improving environmental information disclosure quality(Kong et al., 2022 [21]), mitigating financial mismatch(Li et al., 2023 [25]), improving internal control(Liu et al., 2022 [20]; Ma et al., 2023 [27]), and solving internal and external information constraint(Liu et al., 2022 [20]), but lacks the examination of the role mechanism of digital finance in influencing corporate green innovation, such as stimulating consumer demand and alleviating distortion of the factor market. Second, the existing research on the heterogeneity of the impact of digital finance on enterprise green innovation is only based on the level of regional economic development(Liu et al., 2022 [18]; Li et al., 2022 [23] Rao et al., 2022 [24]), the degree of pollution in the industry(Liu et al., 2022 [18]; Li et al., 2022 [23]), the nature of enterprise property rights(Rao et al., 2022 [24]; Kong et al., 2022 [21]; Li et al., 2022 [23]; Li et al., 2023 [26]), the life cycle of the enterprise(Fan et al., 2022 [19]; Rao et al., 2022 [24]; Xue et al., 2022 [22]), the level of regional financial de-velopment(Li et al., 2023 [26]) and the intensity of regional financial regulation(Li et al., 2023 [26]), etc., and it lacks to analyze the heterogeneity of the impact of digital finance on enterprise green innovation from the perspective of the enterprise scale and whether it has the qualification of high and new technology. This study aims to address these deficiencies by analyzing the influence and mechanisms of digital finance on enterprise green innovation from the perspective of reducing information asymmetry, stimulating consumer demand and alleviating factor market distortion.

3. Comment: The authors should explain what does the star symbols mean in Tables. Moreover, are the figures in paratheses the t-statistics? The authors should not let the readers guess the contents in the tables.

3. Reply:

Thanks for your valuable comment. We have added the meaning of star symbols and the figures in paratheses under Table 3-Table 13(Line 1-2, page 12; Line 1-2, page 13; Line 1-2, page 14; Line 1-2, page 15; Line 1-2, page 17; Line 1-2, page 18; Line 1-2, page 19; Line 11-12, page 20; Line 1-2, page 22; Line 1-2, page 23; Line 1-2, page 24). Modification details are as follows:

Notes: All t-statistics are presented in parentheses under the estimated coefficient. ***, **, and * indicate 1%, 5% and 10% of significance levels, respectively.

4. Comment: The model specification for Section 7 is unclear and I doubt the stepwise mediation regression models are mis-specified in this paper.

4. Reply:

This observation is correct, thank you for pointing this out and we apologize for our mistake. We reconstructed the four-stage stepwise intermediate regression model to verify the mechanism of information asymmetry reduction, consumer demand stimulus and factor market distortion mitigation of digital finance affecting enterprise green innovation(Line 3, page 18 - Line 2, page 23). Modification details are as follows:

1). Mechanism test

1. Mechanism of information asymmetry reduction

In order to verify the mechanism of information asymmetry reduction of digital finance affecting corporate green innovation, with reference to Niu et al.(2023) and combined with the design of benchmark regression model, this paper constructs the following intermediary effect models (6), (7) and (8) :

ASYit=α+β1DIFit+∑φCVit+εit (6)

Yit=α+β1ASYit+∑φCVit+εit (7)

Yit=α+β1DIFit+β2ASYit+∑φCVit+εit (8)

The specific regression results are shown in Table 9. The coefficient of DIF in column (1) is significantly positive, indicating that the development of digital finance is conducive to promoting green innovation of enterprises. The coefficient of DIF in column (2) is significantly negative, indicating that the development of digital finance can reduce information asymmetry; The ASY coefficients in columns (3) and (4) are significantly negative at the 1% level, and the results of stepwise regression method in column (4) show that the DIF coefficient is lower than that in column (1). On this basis, the Sobel test is further conducted in this paper, and it can be found that the Z-value statistic is 13.350, which is significant at 1% level. At the same time, Bootstrap (1000 times) sampling test was conducted in this paper, and it was found that the confidence interval of mediating effect with 95% confidence was [0.0087352, 0.0163977], without 0. The above results indicated that the reduction of information asymmetry played a mediating effect. That is, the development of digital finance will reduce in-formation asymmetry and promote green innovation of enterprises. Hypothesis 2a in this paper is verified.

Table 9. Test results of mediating mechanism: information asymmetry reduction.

 (1) (2) (3) (4)

 PAT ASY PAT PAT

DIF 0.0313*** -0.0009*** 0.0187***

 (7.0883) (-14.1697) (4.1634)

ASY -14.1724*** -13.9329***

 (-7.8536) (-7.6638)

Size 8.1154*** -0.2564*** 4.7640*** 4.5431***

 (12.5762) (-31.5685) (10.7245) (10.4924)

Age -13.6774*** 0.0527*** -11.1946*** -12.9431***

 (-7.6385) (4.7335) (-7.2501) (-7.3819)

lev 0.2798 0.4086*** 5.9037*** 5.9733***

 (0.1584) (15.9915) (4.2492) (4.2677)

Top1 12.2071** 0.5130*** 19.3474*** 19.3542***

 (3.1316) (16.1489) (4.8995) (4.9027)

Mfee 0.0037*** 0.0001 0.0035** 0.0044***

 (3.3422) (0.8571) (2.8453) (3.7408)

Growth 0.4388 -0.0002 0.4361 0.4366

 (1.8886) (-0.0132) (1.6771) (1.7417)

ROE 1.5939* -0.1869*** -1.5422 -1.0097

 (2.1068) (-4.0271) (-1.7856) (-1.2131)

TobinQ -1.7457*** -0.1341*** -3.2494*** -3.6145***

 (-4.6938) (-11.2540) (-7.1932) (-7.5258)

Mshare -22.9440*** 0.1560 -19.4900*** -20.7704***

 (-5.8281) (1.7574) (-4.8799) (-5.0984)

Indep 40.0440*** -0.5546*** 31.7705*** 32.3172***

 (4.1228) (-6.1272) (3.3574) (3.4012)

Dual 2.5448** -0.0207* 2.3310** 2.2564**

 (3.1065) (-2.4813) (3.0127) (2.8891)

SOE 1.2175* 0.0467*** 1.4892* 1.8678**

 (2.0381) (5.5208) (2.3985) (2.9909)

Board -0.4595 -0.0327 -1.7738 -0.9156

 (-0.1909) (-1.1861) (-0.7477) (-0.3888)

Constant -1.6e+02*** 5.6298*** -81.4697*** -77.5154***

 (-11.9723) (36.6899) (-7.8876) (-7.5698)

Sobel Z 13.350***

Bootstrap lower 0.0087352

Bootstrap upper 0.0163977

N 22781 22781 22781 22781

F 23.2996 243.7511 24.5772 23.2654

Adj_R2 0.0696 0.3344 0.0907 0.0912

Notes: All z-statistics are presented in parentheses under the estimated coefficient. ***, **, and * indicate 1%, 5% and 10% of significance levels, respectively.

2. Mechanism of consumer demand stimulus

In order to verify the consumption structure optimization mechanism of digital finance affecting corporate green innovation, this paper constructs the following intermediary effect models (9), (10) and (11) with reference to the design ideas of intermediary effect mentioned above:

structit=α+β1DIFit+∑φCVit+εit (9)

Yit=α+β1structit+∑φCVit+εit (10)

Yit=α+β1DIFit+β2structit+∑φCVit+εit (11)

The specific regression results are shown in Table 10. The coefficient of DIF in column (1) is significantly positive, indicating that the development of digital finance is conducive to promoting green innovation of enterprises. The coefficient of DIF in column (2) is significantly positive, indicating that the development of digital finance can optimize the consumption structure; struct coefficients in columns (3) and (4) are significantly positive at the 1% level, and the results of stepwise regression method in column (4) show that DIF coefficient has decreased compared with column (1). On this basis, the Sobel test is further conducted in this paper, and it can be found that the Z-value statistic is 7.004, which is significant at 1% level. At the same time, Bootstrap (1000 times) sampling test was conducted in this paper, and it was found that the confidence interval of the intermediary effect with 95% confidence was [0.0062394,0.0125897], excluding 0. The above results indicated that the optimization of consumption structure played a mediating effect. That is, the development of digital finance will optimize the consumption structure, thus promoting green innovation of enterprises.

The analysis of the theoretical mechanism above shows that digital finance increases the total amount of residential consumption and optimizes the structure of residential consumption. Column (2) of Table 9 tests the impact of digital finance and its sub-dimensional indicators on the structure of residents' consumption, and the results show that the estimated coefficients of digital finance and its usage depth and digitization level are all significantly positive at the 1% level. Column (3) tests the impact of digital finance and its sub-dimensional indicators on the residents’ consumption levels, and the results show that the estimated coefficients of digital finance and its usage depth and digitization level are significantly positive at the 1% level, the estimated coefficients of its coverage breadth are significantly positive at the 10% level. This indicates that digital finance and its coverage breadth, usage depth and digitalization level significantly increase the total amount of residential consumption; digital finance and its usage depth, and digitalization level significantly optimize the structure of residents' consumption; and Hypothesis 2b is verified.

Table 10. Test results of mediating mechanism: consumption structure optimization.

 (1) (2) (3) (4)

 PAT struct PAT PAT

DIF 0.0186*** 135.4656*** 0.0092***

 (11.1305) (110.9647) (4.3799)

struct 0.0001*** 0.0001***

 (9.9015) (5.7971)

Size 4.1842*** 560.0970*** 4.2197*** 4.1453***

 (20.9486) (7.4748) (21.6741) (20.8302)

Age -5.2247*** -3.9e+03*** -4.3106*** -4.9522***

 (-11.9460) (-17.8263) (-10.3081) (-11.7184)

lev 1.4406** -1.4e+03*** 1.5204** 1.5379**

 (2.9716) (-4.3928) (3.1658) (3.1830)

Top1 1.4172 -3.5e+03*** 1.7045 1.6583

 (1.2939) (-6.5394) (1.5541) (1.5094)

Mfee 0.0008 -2.0399** 0.0007 0.0010*

 (1.8704) (-3.0806) (1.6299) (2.2057)

Growth 0.1270 1.0e+03*** 0.0292 0.0552

 (1.1601) (5.2728) (0.2803) (0.5231)

ROE 1.4586*** 877.8716 1.2230*** 1.3976***

 (3.8778) (1.2402) (3.5045) (3.8720)

TobinQ -1.3231*** -1.4e+02 -1.1858*** -1.3136***

 (-8.2853) (-1.2065) (-7.9283) (-8.1830)

Mshare -8.4127*** 2.7e+03 -8.2830*** -8.6035***

 (-4.4977) (1.7230) (-4.3973) (-4.5745)

Indep 14.9400*** -6.6e+02 14.8730*** 14.9860***

 (5.2012) (-0.4338) (5.1822) (5.2246)

Dual 1.2389*** 1.1e+03*** 1.1614*** 1.1636***

 (4.4963) (6.3788) (4.1865) (4.1957)

SOE -0.0364 484.1998** -0.2042 -0.0701

 (-0.1480) (2.8515) (-0.8338) (-0.2856)

Board 1.0380 -1.1e+03* 0.8754 1.1142

 (1.2378) (-2.4838) (1.0425) (1.3261)

Constant -84.9128*** 4.7e+03* -87.0219*** -85.2428***

 (-19.9255) (2.4621) (-20.9770) (-19.9393)

Sobel Z 7.004***

Bootstrap lower 0.0062394

Bootstrap upper 0.0125897

N 22781 22781 22781 22781

F 70.7043 1.3e+03 62.2080 68.7575

Adj_R2 0.1367 0.4638 0.1379 0.1385

Notes: All z-statistics are presented in parentheses under the estimated coefficient. ***, **, and * indicate 1%, 5% and 10% of significance levels, respectively.

In order to verify the total consumption increased mechanism of digital finance affecting corporate green innovation, this paper constructs the following intermediary effect models (12), (13) and (14) according to the design ideas of intermediary effect mentioned above:

totalit=α+β1DIFit+∑φCVit+εit (12)

Yit=α+β1totalit+∑φCVit+εit (13)

Yit=α+β1DIFit+β2totalit+∑φCVit+εit (14)

The specific regression results are shown in Table 11. The coefficient of DIF in column (1) is significantly positive, indicating that the development of digital finance is conducive to promoting green innovation of enterprises. The coefficient of DIF in column (2) is significantly positive, indicating that the development of digital finance can increase the total consumption; The total coefficients in columns (3) and (4) are significantly positive at the 1% level, and the results of stepwise regression method in column (4) show that the DIF coefficient has decreased compared with that in column (1). On this basis, the Sobel test is further conducted in this paper, and it can be found that the Z-value statistic is 7.798, which is significant at 1% level. At the same time, Bootstrap (1000 times) sampling test was conducted in this paper, and it was found that the confidence interval of the mediating effect with 95% confidence was [0.0071948, 0.0130326], excluding 0. The above results indicated that the increase in total consumption played a mediating effect. That is, the development of digital finance will increase the total consumption and thus promote the green innovation of enterprises. Hypothesis 2b in this paper is verified.

Table 11. Test results of mediating mechanism: total consumption increased.

 (1) (2) (3) (4)

 PAT total PAT PAT

DIF 0.0186*** 81.0888*** 0.0085***

 (11.1305) (110.1486) (4.0808)

total 0.0002*** 0.0001***

 (10.6382) (6.6021)

Size 4.1842*** 322.9818*** 4.2158*** 4.1439***

 (20.9486) (6.9499) (21.6880) (20.8193)

Age -5.2247*** -2.4e+03*** -4.3188*** -4.9233***

 (-11.9460) (-17.2275) (-10.3692) (-11.6535)

lev 1.4406** -9.0e+02*** 1.5358** 1.5524**

 (2.9716) (-4.5432) (3.2039) (3.2190)

Top1 1.4172 -2.0e+03*** 1.7091 1.6722

 (1.2939) (-6.3628) (1.5607) (1.5245)

Mfee 0.0008 -1.6385*** 0.0008 0.0011*

 (1.8704) (-3.6517) (1.8006) (2.3266)

Growth 0.1270 713.5912*** 0.0116 0.0380

 (1.1601) (5.3409) (0.1111) (0.3615)

ROE 1.4586*** 590.7497 1.2184*** 1.3850***

 (3.8778) (1.2399) (3.4868) (3.8473)

TobinQ -1.3231*** -1.2e+02* -1.1854*** -1.3079***

 (-8.2853) (-1.9892) (-7.9907) (-8.1810)

Mshare -8.4127*** 1.1e+03 -8.2149*** -8.5449***

 (-4.4977) (1.1642) (-4.3683) (-4.5504)

Indep 14.9400*** -69.4059 14.8283*** 14.9486***

 (5.2012) (-0.0734) (5.1690) (5.2144)

Dual 1.2389*** 804.8435*** 1.1325*** 1.1386***

 (4.4963) (7.6436) (4.0855) (4.1085)

SOE -0.0364 291.1055** -0.1992 -0.0727

 (-0.1480) (2.7475) (-0.8138) (-0.2963)

Board 1.0380 -7.1e+02** 0.9010 1.1271

 (1.2378) (-2.5923) (1.0732) (1.3416)

Constant -84.9128*** 6.2e+03*** -87.4946*** -85.6904***

 (-19.9255) (5.1605) (-21.0872) (-19.9690)

Sobel Z 7.798***

Bootstrap lower 0.0071948

Bootstrap upper 0.0130326

N 22781 22781 22781 22781

F 70.7043 1.3e+03 62.3932 68.0055

Adj_R2 0.1367 0.4452 0.1384 0.1390

Notes: All z-statistics are presented in parentheses under the estimated coefficient. ***, **, and * indicate 1%, 5% and 10% of significance levels, respectively.

3. Mechanism of factor market distortion mitigation

In order to verify the mitigation mechanism of factor market distortion of digital finance affecting enterprise green innovation, this paper constructs the following intermediary effect models (15), (16) and (17) according to the design ideas of intermediary effect mentioned above:

Distit=α+β1DIFit+∑φCVit+εit (15)

Yit=α+β1Distit+∑φCVit+εit (16)

Yit=α+β1DIFit+β2Distit+∑φCVit+εit (17)

The specific regression results are shown in Table 12. The coefficient of DIF in column (1) is significantly positive, indicating that the development of digital finance is conducive to promoting green innovation of enterprises. The coefficient of DIF in column (2) is significantly negative, indicating that the development of digital finance can alleviate factor market distortion; Dist coefficients in columns (3) and (4) are significantly negative at the 1% level, and the results of stepwise regression method in column (4) show that DIF coefficient is lower than that in column (1). On this basis, the Sobel test is further conducted in this paper, and it can be found that the Z-value statistic is 3.608, which is significant at 1% level. At the same time, Bootstrap (1000 times) sampling test was conducted in this paper, and it was found that the confidence interval of mediating effect with 95% confidence was [0.0005324, 0.0017878], excluding 0. The above results indicated that mitigation of factor market distortion played a mediating effect. In other words, the development of digital finance will alleviate the distortion of factor market, thus promoting the green innovation of enterprises. The hypothesis 2c in this paper is verified.

The impact of digital finance on factor market distortions is further discussed in this study. As per Hypothesis 2c, digital finance has the potential to alleviate factor market distortions. Column (4) of Table 9 presents the estimated outcomes of the impact of digital finance and its sub-dimensional indicators on factor market distortions. The results suggest a significantly positive coefficient for digital finance and its digitization level at the 1% level. These findings demonstrate that digital finance, along with its digitization level, plays a crucial role in reducing and mitigating factor market distortions, thereby promoting enterprise green innovation, and Hypothesis 2c is verified.

Table 12. Test results of mediating mechanism: factor market distortion mitigation.

 (1) (2) (3) (4)

 PAT Dist PAT PAT

DIF 0.0313*** -0.0117*** 0.0301***

 (7.0883) (-8.4393) (7.0359)

Dist -0.1074*** -0.0988***

 (-4.0957) (-3.8467)

Size 8.1154*** -0.1563* 8.5569*** 8.1000***

 (12.5762) (-1.9902) (13.0801) (12.5693)

Age -13.6774*** -1.7861*** -11.0515*** -13.8538***

 (-7.6385) (-6.1383) (-6.8627) (-7.5919)

lev 0.2798 1.8678*** 0.2084 0.4643

 (0.1584) (4.6870) (0.1185) (0.2614)

Top1 12.2071** 2.5060*** 12.2657** 12.4547**

 (3.1316) (4.1302) (3.1363) (3.1774)

Mfee 0.0037*** -0.0037*** 0.0018 0.0033**

 (3.3422) (-5.0224) (1.6904) (3.1670)

Growth 0.4388 -0.9792*** 0.3329 0.3421

 (1.8886) (-6.4549) (1.5144) (1.5502)

ROE 1.5939* 0.6982 0.8772 1.6629*

 (2.1068) (1.3792) (1.2017) (2.2572)

TobinQ -1.7457*** -0.1407 -1.1159*** -1.7596***

 (-4.6938) (-1.3637) (-3.4654) (-4.7340)

Mshare -22.9440*** -3.7953* -21.3338*** -23.3189***

 (-5.8281) (-2.4277) (-5.5405) (-5.8691)

Indep 40.0440*** 0.9431 39.4736*** 40.1372***

 (4.1228) (0.5288) (4.0709) (4.1306)

Dual 2.5448** 1.0256*** 2.7840*** 2.6461**

 (3.1065) (4.8702) (3.4510) (3.2638)

SOE 1.2175* -2.9981*** 0.2628 0.9213

 (2.0381) (-15.8264) (0.4668) (1.5801)

Board -0.4595 1.4340** -1.6860 -0.3179

 (-0.1909) (2.7759) (-0.6898) (-0.1316)

Constant -1.6e+02*** 21.5859*** -1.6e+02*** -1.5e+02***

 (-11.9723) (10.3468) (-12.5865) (-11.9501)

Sobel Z 3.608***

Bootstrap lower 0.0005324

Bootstrap upper 0.0017878

N 22781 22781 22781 22781

F 23.2996 62.4946 24.4411 23.8649

Adj_R2 0.0696 0.0352 0.0688 0.0702

Notes: All z-statistics are presented in parentheses under the estimated coefficient. ***, **, and * indicate 1%, 5% and 10% of significance levels, respectively.

Reviewer#2:

1. Comment: Title should be redesigned. The role of digital finance in driving green innovation… is very huge, it is not precisely summarized what you have done.

1. Reply:

Thank you for your positive comments and valuable suggestions to improve the quality of our manuscript. After discussion, we decided to change the title to “Impact of digital finance on enterprise green innovation: from the perspective of information asymmetry, consumer demand and factor market distortions” (Line 2-5, page 1)

2. Comment: Why the analysis is limited by 2020 year? I suppose that 2021 data is already available in mentioned sources.

2. Reply: 

We thank the reviewer for pointing this out. We supplemented the data for 2021 and re-exported and reinterpreted the results of descriptive statistics and regression(Line 21, page 6 - Line 2, page 24).

3. Comment: Research gaps and contribution should be well mentioned in the introduction. A good research gap can give the reader more insight.

3. Reply: 

We are very grateful to the reviewers for pointing this out. We have added a separate paragraph in the introduction (Line 18, page 2 - Line 41, page 2) to clarify research gaps and contribution. Modification details are as follows:

1). the main contribution in the introduction

The main contributions of this study to the existing literature are summarized as follows: First, most existing related literatures have explored the mechanism of digital finance on enterprise green innovation from the perspectives of alleviating financing constraint(Liu et al., 2022; Fan et al., 2022; Liu et al., 2022; Kong et al., 2022; Xue et al., 2022; Li et al., 2022; Li et al., 2023; Ma et al., 2023), improving corporate transparency(Rao et al.,2022), increasing R&D investment(Liu et al., 2022; Li et al., 2023), enhancing financial flexibility(Fan et al., 2022), improving environmental information disclosure quality(Kong et al., 2022), mitigating financial mismatch(Li et al., 2023), improving internal control(Liu et al., 2022; Ma et al., 2023), and solving internal and external information constraint(Liu et al., 2022), whereas our study theoretically analyzes and empirically examines the mechanisms of information asymmetry re-duction, consumer demand stimulus and factor market distortion mitigation by which digital finance affects enterprise green innovation, as well as the impacts of intellectual property protection and environmental governance on the relationship between digital finance and enterprise green innovation, which sheds new light on digital finance influencing enterprise green innovation. Second, most existing studies on the heterogeneity of the impact of digital finance on enterprise green innovation are based on the level of regional economic development(Liu et al., 2022; Li et al., 2022; Rao et al., 2022), the degree of pollution in the industry(Liu et al., 2022; Li et al., 2022), the nature of enterprise property rights(Rao et al., 2022; Kong et al., 2022; Li et al., 2022; Li et al., 2023), the life cycle of the enterprise(Fan et al., 2022; Rao et al., 2022; Xue et al., 2022), the level of regional financial development(Li et al., 2023) and the intensity of regional financial regulation(Li et al., 2023), whereas our study analyzes the heterogeneity of the results of the study from the perspectives of enterprise scale and whether they possess high-tech qualifications, and enriches the study on the impact of digital finance on the green innovation of enterprises with different characteristics.

4. Comment: The descriptive statistics should check for normality and for the asymmetry of the distributions of the variables. Distortions can be calculated using skewness and kurtosis( Min=median=0 for many variables).

4. Reply: 

Thanks for your valuable comment. We have added skewness and kurtosis in the descriptive statistics section to check the normality and asymmetry of the variable distribution(Line 12, page 9 - Line 1, page 10). Modification details are as follows:

1). Descriptive statistics

The descriptive statistics of the variables are displayed in Table 2, which reveal noteworthy findings. Specifically, PAT has a mean of 8.318, with a standard deviation of 48.44, a minimum value of 0, and a maximum value of 1612, indicating significant variations in green innovation among Chinese enterprises. The skewness of PAT is greater than 18, showing a typical right-skewness distribution, and the kurtosis is greater than 441, indicating that the distribution is peak-like compared with the normal distribution. INPAT and UPAT show the same characteristics. In addition, DIF has a mean of 213.1, a standard deviation of 77.12, a minimum value of 21.26, and a maximum value of 359.7 suggesting that the level of digital finance development across regions in China is relatively uneven, showing polarization. The skewness of DIF is close to 0, and the kurtosis is less than 3, indicating that the distribution of DIF is flat compared with the normal distribution without obvious skewness. Its coverage breadth, usage depth, and digitization level show the same characteristics.

Table 2. Descriptive statistics.

Variable N Mean Std. Dev. Min Median Max Skewness Kurtosis

PAT 22781 8.318 48.44 0 0 1612 18.31 441.4

INPAT 22781 4.876 33.53 0 0 1381 21.43 605.1

UPAT 22781 3.442 19.02 0 0 709 19.09 488.9

DIF 22781 213.1 77.12 21.26 222.2 359.7 -0.313 2.189

DIFB 22781 213.2 76.08 -10.49 219.8 371.8 -0.212 2.343

DIFD 22781 208.5 78.32 12.49 216.1 354.3 -0.220 1.994

DIFL 22781 221.3 90.41 3.390 246.8 581.2 -0.670 2.313

Size 22781 22.39 1.398 12.24 22.23 48.31 1.182 14.01

Age 22781 2.914 0.350 0.693 2.996 3.829 -1.086 4.968

lev 22781 0.418 0.254 -1.855 0.421 9.429 2.648 83.88

Top1 22781 0.0920 0.166 0.009 0.009 0.961 1.985 6.055

Mfee 22781 0.211 14.02 0 0.0730 2115 150.5 22700

Growth 22781 -0.298 0.903 -13.39 -0.129 57.41 15.77 824.7

ROE 22781 0.0490 0.231 -7.016 0.0640 14.02 7.325 702.6

TobinQ 22781 0.213 0.838 0 0.005 25.51 8.657 132.0

Mshare 22781 0.00800 0.0480 0 0 0.692 7.737 70.38

Indep 22781 0.375 0.0560 0 0.354 0.810 1.250 6.493

Dual 22781 0.232 0.422 0 0 1 1.268 2.607

SOE 22781 0.430 0.495 0 0 1 0.284 1.081

Board 22781 2.141 0.203 0 2.197 3.819 -0.338 5.310

5. Comment: The interpretation of the tables3, 4 is superficial. You only mention the results sign and significance and the validation of the hypothesis. What does it mean for the Chinese A-share listed firms ?what are the implications for Chinese government policy? Does it have high standard implication for Chinese economy?

5. Reply: 

We are very grateful to you for pointing out this valuable and very central issue, which is important for refining and improving the quality of the whole article. We have enriched our interpretation of Table 3,4(Line 1, page 10 - Line 2, page 13). Modification details are as follows:

1). Analysis of the results of the return

1. Benchmark regression results

Table 3 displays the outcomes of the benchmark regression analysis conducted to examine the association between digital finance and green innovation in enterprises. In models (1) to (3), only the "time–province" fixed effects were controlled, and the results showed that the impact of digital finance development (DIF) on enterprise green innovation was examined through a regression analysis, which revealed positive coefficients (βPAT=0.2412, βINPAT=0.2697, βUPAT=0.1054) for the total level of green innovation (PAT), green invention innovation (INPAT), and green utility model innovation (UPAT). All coefficients were statistically significant at the 1% level. After including the relevant control variable set (models (4) to (6)), the statistical significance level of digital finance development (DIF) on the total green innovation level of enterprises (PAT) and green utility model innovation (UPAT) regression decreased. The research results show that under the influence of digital finance, enterprises' green innovation ability is gradually strengthened. With the support of digital finance, enterprises have improved their ability to collect, integrate and analyze information, which can help enterprises judge the status of green innovation and market potential, and improve the effectiveness of green innovation decisions of enterprises. In addition, under the supervision pressure from outside the market, enterprises will pay more attention to how to improve the core innovation competitiveness, so as to concentrate resources on such green invention innovation activities with high gold content, and the promotion effect is weak for those patent innovations with low economic potential. The abovementioned results prove Hypothesis 1.

Table 3. The impact of digital finance on enterprise green innovation: benchmark regression.

 (1) (2) (3) (4) (5) (6)

 PAT INPAT UPAT PAT INPAT UPAT

DIF 0.2412*** 0.2697*** 0.1054*** 0.1412** 0.2053*** 0.0486

 (3.3948) (3.7023) (2.9687) (2.2559) (3.2001) (1.5843)

Size 19.0160*** 15.5699*** 8.2557***

 (4.9287) (4.2377) (5.3600)

Age -29.6241*** -20.7352*** -14.6130***

 (-3.7899) (-3.6727) (-3.6221)

lev -2.3339 -8.6100 7.4236*

 (-0.2669) (-1.1429) (1.9093)

Top1 6.6467 -8.9407 15.4544**

 (0.3963) (-0.6518) (2.0009)

Mfee -12.1058** -5.2157 -8.1610**

 (-2.2065) (-1.5865) (-2.3993)

Growth 0.5067 0.6769 -0.3611

 (0.4767) (0.7370) (-0.6745)

ROE 5.6204 4.2247 4.6433**

 (1.5890) (1.4180) (2.4021)

TobinQ 3.3534 2.3588 1.1433

 (1.0608) (1.1141) (0.5866)

Mshare -1.9191 7.9416 -5.1134

 (-0.1018) (0.5649) (-0.4311)

Indep 43.2382 27.8085 23.7394

 (1.4495) (1.3230) (1.4632)

Dual 4.8229* 4.6548** 1.1462

 (1.7508) (2.1169) (0.8807)

SOE 1.5075 2.7335 -0.7720

 (0.5505) (1.1431) (-0.6119)

Board 4.5568 8.3589 -1.8162

 (0.5237) (1.0525) (-0.5076)

year Yes Yes Yes Yes Yes Yes

province Yes Yes Yes Yes Yes Yes

N 22781 22781 22781 22781 22781 22781

PseudoR2 0.0128 0.0141 0.0190 0.0310 0.0347 0.0451

Notes: All z-statistics are presented in parentheses under the estimated coefficient. ***, **, and * indicate 1%, 5% and 10% of significance levels, respectively.

In order to provide a more precise representation of the impact of digital finance on enterprise green innovation, this research has classified the digital finance index into three distinct levels: the coverage breadth, usage depth and digitization level. Based on this, this study analyzed which dimensions of digital finance development can significantly promote enterprise green innovation. The results in Table 4 show the impact of the development of the dimension of "coverage breadth - usage depth - digitization level" on enterprises' green in-novation activities: the regression coefficient of digital financial coverage breadth (DIFB) on enterprises' green invention and innovation (βINPAT=0.1549) is positive, passing the 1% statistical significance test. The regression coefficients of total green innovation level and green utility model innovation (βPAT=0.1136 and βUPAT=0.0439) were positive, and the significance decreased. The usage depth of digital finance (DIFD) only had a significant effect on green invention innovation (βINPAT=0.1567). The digitization level of digital finance (DIFL) has no significant impact on enterprises' green innovation. This shows that the development of digital finance mainly stimulates the green innovation of enterprises by ex-panding the coverage and deep mining. The coverage breadth of digital finance can better reflect the fairness of digital finance, so that small and medium-sized enterprises can reach financial services, and reduce the uneven allocation of financial resources. The usage depth of digital finance reflects the application results of digital finance, and describes the specific financial functions of digital finance in the business activities of enterprises. Digitization level of digital finance, as the embodiment of the low threshold and low cost characteristics of digital finance, can increase the demand for financial services, but compared with the breadth and depth of digital financial applications, its green innovation effect is relatively small. If the development of digital finance only relies on digitalization without achieving extensive coverage and deep mining, it is difficult to provide support for micro-economic entities, nor can it provide sustained impetus for high-quality economic development.

Table 4. The impact of digital finance development on enterprise green innovation: indicator dimensionality reduction.

 (1) (2) (3)

 PAT INPAT UPAT

DIFB 0.1136** 0.1549*** 0.0439*

 (2.3816) (3.1593) (1.8865)

Control Yes Yes Yes

year Yes Yes Yes

province Yes Yes Yes

N 22781 22781 22781

PseudoR2 0.0311 0.0347 0.0452

DIFD 0.0917 0.1567*** 0.0242

 (1.5471) (2.7426) (0.8178)

Control Yes Yes Yes

year Yes Yes Yes

province Yes Yes Yes

N 22781 22781 22781

PseudoR2 0.0310 0.0345 0.0451

DIFL 0.0071 0.0354 -0.0136

 (0.2329) (1.3041) (-0.9605)

Control Yes Yes Yes

year Yes Yes Yes

province Yes Yes Yes

N 22781 22781 22781

PseudoR2 0.0309 0.0343 0.0451

Notes: All z-statistics are presented in parentheses under the estimated coefficient. ***, **, and * indicate 1%, 5% and 10% of significance levels, respectively.

6. Comment: Useless tests of robustness via independent variable lag.(page 10). For example you can select the lag using schwartz or Akaike information criterion.

6. Reply: 

Thanks for your valuable comment. After consideration, we have decided to remove the robustness test through independent variable lag(Line 3, page 14 - Line 2, page 15). Modification details are as follows:

1). Robustness test

1. Addition of control variables

This study incorporated macro-level data on various factors, including the share of tertiary industry value added in GDP, expenditure on science and education as a percentage of total regional fiscal expenditure, foreign direct investment as a percentage of GDP, government subsidies received by enterprises, and marketization index of prefecture-level cities. These factors were added to the model, and the regression results are presented in column (1) through column (3) of Table 6. The findings indicate that the regression coefficient for digital finance retains its statistical significance with a positive value, indicating that the benchmark regression results are generally robust.

2. Replacement regression model

To conduct the analysis, a bidirectional fixed-effect model was utilized. The regression results are presented in columns (4) through (6) of Table 6, and the regression coefficient for digital finance remains significantly positive. This finding is consistent with the previous conclusion.

3. Exclusion of some data

On the one hand, it is noteworthy that the 2015 Chinese stock market crash may have had an impact on both the development of digital finance and the green innovation behavior of enterprises, and this study excluded the 2015 data; on the other hand, due to the large economic specificity of the municipalities in China, it is also possible that there are differences in the development of digital finance and the green innovation activities among enterprises., thus, this study excluded the sample data of the municipalities and re-ran the regression. The regression results are shown in columns (7) - (12) in Table 6. The positive statistical significance of the regression coefficient for digital finance persists, and the results are still robust.

4. Replacement of the independent variable

The value of a green patent and the quality of green innovation for an enterprise that has applied for the patent may be positively correlated with the frequency of citations of the patent. To evaluate the quality of green innovation among enterprises, this study employed the number of green patents cited (GPR) as a measure. Additionally, the research examined how digital finance and its sub-dimensions influence ecological innovation in enterprises. The regression analysis results are presented in columns (13) through (16) of Table 6, and they confirm that the positive effect of digital finance on green innovation is still significant. This further supports the robustness of the benchmark regression results.

Table 6. Robustness test results.

 Increased control variables Replacement Regression model

 (1) (2) (3) (4) (5) (6)

 PAT INPAT UPAT PAT INPAT UPAT

DIF 0.1513** 0.2130*** 0.0483 0.0778*** 0.0653*** 0.0126

 (2.5074) (3.5005) (1.5579) (2.7056) (2.8491) (1.2562)

Control Yes Yes Yes Yes Yes Yes

year Yes Yes Yes Yes Yes Yes

province Yes Yes Yes Yes Yes Yes

N 22781 22781 22781 22781 22781 22781

PseudoR2 0.0354 0.0398 0.0476 

 Excluding 2015 Excluding municipalities

 (7) (8) (9) (10) (11) (12)

 PAT INPAT UPAT PAT INPAT UPAT

DIF 0.1559** 0.2134*** 0.0612** 0.1306** 0.1592*** 0.0469**

 (2.4811) (3.3093) (2.0141) (2.5627) (3.1375) (2.0733)

Control Yes Yes Yes Yes Yes Yes

year Yes Yes Yes Yes Yes Yes

province Yes Yes Yes Yes Yes Yes

N 20710 20710 20710 18271 18271 18271

PseudoR2 0.0315 0.0353 0.0463 0.0306 0.0351 0.0440

 Replaced Independent variable 

 (13) (14) (15) (16) 

 GPR GPR GPR GPR 

DIF 1.5224** 

 (2.0931) 

DIFB 1.1562** 

 (2.0790) 

DIFD 1.1198* 

 (1.7807) 

DIFL 0.3041 

 (1.1499) 

Control Yes Yes Yes Yes 

year Yes Yes Yes Yes 

province Yes Yes Yes Yes 

N 22781 22781 22781 22781 

PseudoR2 0.0169 0.0169 0.0168 0.0168 

Notes: All z-statistics are presented in parentheses under the estimated coefficient. ***, **, and * indicate 1%, 5% and 10% of significance levels, respectively.

7. Comment: The heterogeneity analysis results discussions aren’t based on previous studies. Why SMEs green innovations are more impacted by the digital finance while that impact isn’t significant for large enterprises?

7. Reply: 

We are very grateful to you for pointing out this valuable and very central issue, which is important for refining and improving the quality of the whole article. Based on previous studies, we add an analysis of the reasons why digital finance only affects SMEs and high-tech enterprises (Line 3, page 15 - Line 2, page 18). Modification details are as follows:

1). Heterogeneity analysis

1. Sub-sample study based on firm size

This study divided the sample of enterprises according to their total assets (enterprises with total assets above the mean are large enterprises; otherwise, they are SMEs) and examined the variability of the impact of digital finance on enterprises of different sizes. The results, presented in Table 7, demonstrate the significantly positive impact of both digital finance and its coverage breadth and usage depth indicators on SMEs' green innovation, green invention innovation and green utility model innovation of enterprises, the regression coefficients for digital financial and digitization level indicators are significantly positive only in relation to green innovation and green invention innovation among SMEs. However, the regression coefficients of digital finance and its sub-dimension indicators do not show significant effects on green innovation among large enterprises.

This may be because SMEs are more likely to be excluded from the threshold of traditional financial services due to their shortage of mortgage resources, high operational risks, and relatively imperfect credit records and have very limited financing channels, while due to the inclusiveness of digital finance, SMEs are more motivated to use them to obtain green innovation financing. However, large enterprises have relatively sufficient funds and face fewer financing constraints, and digital finance has less impact on them.

Table 7. Sample results of enterprise size.

 SMEs Big enterprises

 (1) (2) (3) (4) (5) (6)

 PAT INPAT UPAT PAT INPAT UPAT

DIF 0.0900*** 0.0993*** 0.0450*** 0.0774 0.1074 0.0005

 (3.9953) (5.2167) (2.6851) (0.6532) (1.0253) (0.0094)

Control Yes Yes Yes Yes Yes Yes

year Yes Yes Yes Yes Yes Yes

province Yes Yes Yes Yes Yes Yes

N 12397 12397 12397 10384 10384 10384

PseudoR2 0.0380 0.0412 0.0444 0.0249 0.0262 0.0369

DIFB 0.0658*** 0.0714*** 0.0341*** 0.0705 0.0848 0.0115

 (3.9989) (5.1803) (2.8310) (0.7697) (1.0530) (0.2711)

Control Yes Yes Yes Yes Yes Yes

year Yes Yes Yes Yes Yes Yes

province Yes Yes Yes Yes Yes Yes

N 12397 12397 12397 10384 10384 10384

PseudoR2 0.0380 0.0412 0.0445 0.0249 0.0262 0.0369

DIFD 0.0689*** 0.0816*** 0.0312* 0.0442 0.0770 -0.0125

 (3.0801) (4.4065) (1.7665) (0.4073) (0.8122) (-0.2483)

Control Yes Yes Yes Yes Yes Yes

year Yes Yes Yes Yes Yes Yes

province Yes Yes Yes Yes Yes Yes

N 12397 12397 12397 10384 10384 10384

PseudoR2 0.0372 0.0398 0.0439 0.0249 0.0262 0.0369

DIFL 0.0229** 0.0264*** 0.0072 -0.0159 0.0126 -0.0293

 (2.3163) (2.9670) (0.9498) (-0.3085) (0.2822) (-1.2678)

Control Yes Yes Yes Yes Yes Yes

year Yes Yes Yes Yes Yes Yes

province Yes Yes Yes Yes Yes Yes

N 12397 12397 12397 10384 10384 10384

PseudoR2 0.0367 0.0383 0.0436 0.0249 0.0262 0.0369

Notes: All z-statistics are presented in parentheses under the estimated coefficient. ***, **, and * indicate 1%, 5% and 10% of significance levels, respectively.

2. Sub-sample research based on high-tech/non-high-tech industries

The study categorized enterprises into high-tech and non-high-tech industries by referring to the high-tech enterprise recognition announcement and re-examination announcement published in the WIND database. Table 8 displays the regression outcomes of the two subgroups. The results indicate that the digital finance and its coverage breadth, usage depth and digitalization level indicators have a significantly positive effect on green innovation, green invention innovation, and green utility model innovation of high-tech industry enterprises. Conversely, the regression analyses fail to confirm the significance of the impact of digital finance and its sub-dimensional indicators on green innovation of non-high-tech industry enterprises.

This may be because compared with non-high-tech enterprises, high-tech enterprises have stronger motivation to finance green innovation projects due to their knowledge-intensive and environment-friendly characteristics. Specifically, green innovation projects account for a large proportion of the income of high-tech enterprises, and green innovation research and development itself is characterized by high investment, high risk and long duration, which makes it difficult for enterprises to meet the capital demand of innovation projects only by internal financing, and also makes enterprises face high external financing costs. However, the characteristics of innovative project financing of high-tech enterprises are contrary to the principle of "liquidity, security and profitability" adhered to by traditional financial institutions, so the uncertainty of credit availability of high-tech enterprises is higher, and the demand for the new financial model of digital inclusive finance is stronger.

Table 8. Sample results of high-tech/non-high-tech industries.

 Non-high-tech industries High-tech industries

 (1) (2) (3) (4) (5) (6)

 PAT INPAT UPAT PAT INPAT UPAT

DIF -0.0876 -0.0003 -0.0544 0.3459*** 0.3703*** 0.1219***

 (-0.9011) (-0.0034) (-1.0374) (4.1251) (3.6239) (5.3596)

Control Yes Yes Yes Yes Yes Yes

year Yes Yes Yes Yes Yes Yes

province Yes Yes Yes Yes Yes Yes

N 13816 13816 13816 8965 8965 8965

PseudoR2 0.0384 0.0432 0.0540 0.0411 0.0429 0.0662

DIFB -0.0567 0.0058 -0.0351 0.2442*** 0.2563*** 0.0906***

 (-0.7652) (0.0929) (-0.8731) (4.2962) (3.7491) (5.3653)

Control Yes Yes Yes Yes Yes Yes

year Yes Yes Yes Yes Yes Yes

province Yes Yes Yes Yes Yes Yes

N 13816 13816 13816 8965 8965 8965

PseudoR2 0.0384 0.0432 0.0540 0.0411 0.0427 0.0665

DIFD -0.0851 0.0059 -0.0524 0.2707*** 0.2973*** 0.0860***

 (-0.8775) (0.0726) (-1.0321) (3.5257) (3.2705) (4.0548)

Control Yes Yes Yes Yes Yes Yes

year Yes Yes Yes Yes Yes Yes

province Yes Yes Yes Yes Yes Yes

N 13816 13816 13816 8965 8965 8965

PseudoR2 0.0384 0.0432 0.0540 0.0399 0.0412 0.0647

DIFL -0.0392 -0.0317 -0.0245 0.1129** 0.1454** 0.0209

 (-0.8369) (-0.7504) (-1.0401) (2.1781) (2.4444) (1.5637)

Control Yes Yes Yes Yes Yes Yes

year Yes Yes Yes Yes Yes Yes

province Yes Yes Yes Yes Yes Yes

N 13816 13816 13816 8965 8965 8965

PseudoR2 0.0384 0.0432 0.0540 0.0390 0.0398 0.0638

Notes: All z-statistics are presented in parentheses under the estimated coefficient. ***, **, and * indicate 1%, 5% and 10% of significance levels, respectively.

8. Comment: How you compare and contrast your research findings with other studies?

8. Reply: 

We thank the reviewer for pointing this out. We have compared and contrasted our research findings with those of other studies in the conclusion section(Line 4, page 24 - Line 41, page 24). Modification details are as follows:

1). Conclusions

In this study, the theoretical mechanism of how digital finance influences enterprise green innovation was systematically analyzed. We used panel data from 2071 A-share listed companies in China from 2011 to 2021 to empirically evaluate the impact of digital finance and its coverage breadth, usage depth, and digitization level on enterprise green innovation. Meanwhile, the heterogeneous effects of digital finance on green innovation of enterprises with different characteristics are investigated. Furthermore, the mechanisms of information asymmetry reduction, consumer demand stimulus, and factor market distortion mitigation by which digital finance affects enterprise green innovation and the impact of intellectual property protection and environmental governance on the relationship between digital finance and enterprise green innovation were further investigated. The main conclusions of this paper are as follows:

First, the development of digital finance has a significant role in promoting the green innovation of enterprises, which is specifically manifested in encouraging the green invention innovation and green utility model innovation of enterprises. Compared with green utility model innovation, digital finance has a stronger incentive effect on green invention and in-novation. The coverage of digital finance has a significant positive impact on enterprise green innovation, enterprise green invention innovation and enterprise green utility model innovation. The depth of use of digital finance only has a significant impact on enterprises' green invention and innovation; The digitalization level of digital finance has no significant impact on enterprises' green innovation. These results are different from those of some scholars, such as Li et al. (2023). They believe that the promotion of green innovation by the development of digital inclusive finance is mainly driven by the depth of application of digital inclusive finance and the digitalization of inclusive finance. These results are also similar to the studies of some scholars, such as Fan et al. (2022) found that the coverage and depth of digital finance can promote the green innovation of enterprises, and the degree of digitalization has no significant impact on the green technology innovation of enterprises, but it lacks the fractal dimension of digital finance.

Second, the impact of digital finance on promoting green innovation of enterprises is heterogeneous. The development of digital finance and its coverage breadth and use depth only has a significant impact on the green innovation, green invention innovation and green utility model innovation of small and medium-sized enterprises and high-tech enterprises. In addition, the digitalization level of digital finance only has a significant promoting effect on the green innovation and green invention innovation of small and medium-sized enterprises and high-tech enterprises. The existing literature lacks the heterogeneity research on green innovation of enterprises with different scale of influence of digital finance and whether they have high-tech qualifications.

Third, digital finance promotes corporate green innovation by reducing information asymmetry, stimulating consumer demand, and alleviating regional factor market distortions. By strengthening intellectual property protection and environmental governance, the role of digital finance in promoting green innovation can be further strengthened.

9. Comment: Section VIII lacks enough policy implications. The policy suggestions are also lack of enough practicability. Authors should put forward detailed policy suggestions which apply to China and to many countries, especially emerging countries.

9. Reply: 

We gratefully appreciate your pointing out the inadequate elaboration of the policy implications section, which is important for us to further improve the study. We have made more specific policy implications for China and other emerging economies, and financial institutions (Line 7, page 25 - Line 37, page 25). Modification details are as follows:

1). Policy implications

According to this research, we promote the following policy implications. From the national perspective, China and other emerging economies should increase their investment in digital infrastructure and, with the help of big data, artificial intelligence, cloud computing and other digital technologies, set up green innovation databases and assessment systems to collect, monitor, calculate and analyse real-time information on the green innovation activities of enterprises, so as to provide financial institutions with the relevant information needed to assess the green innovation capacity and environmental performance of enterprises, so that financial institutions can better support enterprises' green innovations in terms of financing and investment. Meanwhile, China and other emerging economies should promote resource integration and cooperation among enterprises, financial institutions, research institutes and other parties, and set up a collaborative mechanism to jointly research and develop green patented technologies, promote green digital financial products, and carry out demonstration projects, so as to realize optimal allocation of resources and collaborative innovation. Finally, China and other emerging economies should comprehensively promote the marketization process, give full play to the decisive role of the market mechanism in resource allocation, reduce biased financial policies, provide higher-quality digital financial services for green innovation and development, and guide enterprises and scientific research institutes to invest their capital elements in green innovation and R&D activities.

From the perspective of financial institutions, financial institutions should rely on digital finance to realize the digital function of financial infrastructure, use digital technology to strengthen the intelligent identification capability of green enterprises and green projects, and actively provide enterprises with diversified financial products and services such as green credit, green bonds, etc. At the same time, digital financial platforms should guide residents to form a green consumption system through the intelligent recommendation of green products, personalized service of green consumption assessment report, and provision of green con-sumption. Meanwhile, digital financial platforms should guide residents to form green con-sumption concepts and low-carbon lifestyles through the intelligent recommendation of green products, personalized services of green consumption assessment reports, and the provision of green consumption credits, thus encouraging enterprises to accelerate green innovation to meet consumer demand. Finally, Financial institutions should develop differentiated digital financial products and provide personalized digital financial services for enterprises of dif-ferent sizes and industries.

10. Comment: The possible future research directions should be added at the end of the paper, so that in the future the researchers who are interested in this important topic could carry out follow-up studies.

10. Reply: 

Thank you for your positive comments and valuable suggestions to improve the quality of our manuscript. We have added possible future research directions in Part IX so that in the future the researchers who are interested in this important topic could carry out follow-up studies(Line 12, page 26 - Line 28, page 26). Modification details are as follows:

1). Limitations and future research

This paper also has some limitations that warrant further research in the future. First, this study only explores the impact of digital finance on corporate green innovation from the perspectives of reducing information asymmetry, stimulating consumer demand, and mitigating factor market distortions, and future research can explore the realization path of digital finance on corporate green innovation from other perspectives, such as technology spillovers. Second, due to the limitation of data sources, this study only tested the impact of digital finance on corporate green innovation during the period of 2011-2021, and future researchers can extend the timeframe of the study to test the relationship between the two in a longer timeframe. Third, the study samples are all Chinese listed companies, and a large number of Chinese non-listed companies and companies from other countries are not included in the review, and the role of digital finance in promoting green innovation for companies from other countries or Chinese non-listed SMEs needs to be further explored in the future. Finally, the conclusions of this paper come from the quantitative analysis of a large amount of data, and it is difficult to conduct in-depth research on the evolutionary process between variables as in case studies. In the future, researchers can conduct detailed case studies on the green innovation activities of different types of enterprises to reveal the evolutionary process of digital finance affecting the green innovation of enterprises.

11. Comment: Text requires polishing.

11. Reply: 

Thanks very much for your comments. We have polished our paper. Please see if the revised version met the English presentation standard.

12. Comment: Typing repetition p6 (environment protection).

11. Reply: 

We thank the reviewer for pointing this out. We have removed the repetition of "environment protection"(Line 17, page 8). Modification details are as follows:

Drawing on Chen et al. (2016), this study used the frequency of words related to the term "environmental protection" in municipal government work reports (specifically: environmental protection, pollution, energy consumption, emission reduction, emissions, ecology, green, low-carbon, air, chemical oxygen demand, sulfur dioxide, carbon dioxide, PM10, and PM2.5, etc.) as a proxy for environmental governance.

Thank you very much for your attention and time. Look forward to hearing from you.

Yours sincerely.

Zhongan Zhang

August 13, 2023

School of Economics and Management, Xinjiang University

E-mail: 20200500214@stu.xju.edu.cn

---

## [Decision Letter · Decision Letter 1]

30 Nov 2023

Impact of digital finance on enterprise green innovation: From the perspective of information asymmetry, consumer demand and factor market distortions

PONE-D-23-15681R1

Dear Dr. Zhang,

We’re pleased to inform you that your manuscript has been judged scientifically suitable for publication and will be formally accepted for publication once it meets all outstanding technical requirements.

Kind regards,

Zakaria Boulanouar, PhD

Academic Editor

PLOS ONE

Additional Editor Comments (optional):

Reviewers' comments:

Reviewer's Responses to Questions

**Comments to the Author**

1. If the authors have adequately addressed your comments raised in a previous round of review and you feel that this manuscript is now acceptable for publication, you may indicate that here to bypass the “Comments to the Author” section, enter your conflict of interest statement in the “Confidential to Editor” section, and submit your "Accept" recommendation.

Reviewer #1: All comments have been addressed

Reviewer #2: All comments have been addressed

2. Is the manuscript technically sound, and do the data support the conclusions?

Reviewer #1: Yes

Reviewer #2: Yes

3. Has the statistical analysis been performed appropriately and rigorously? 

Reviewer #1: Yes

Reviewer #2: Yes

4. Have the authors made all data underlying the findings in their manuscript fully available?

Reviewer #1: Yes

Reviewer #2: Yes

5. Is the manuscript presented in an intelligible fashion and written in standard English?

Reviewer #1: Yes

Reviewer #2: Yes

6. Review Comments to the Author

Reviewer #1: I would appreciate the great efforts that the authors made to revise the manuscript. I have no further comments to this paper. I think it could be accepted at the present form

Reviewer #2: The author(s) have clearly responded to all my concerns raised in the initial review, resulting in a well-executed manuscript that i believe, it will make a significant contribution to the discourse on the effect of digital finance on the enterprise green innovation. I found the author(s) have satisfactory addressed my previous concerns. They have supplemented the data for 2021. The revised manuscript now includes a more comprehensive discussion of the research gap and the contributions. The author(s) have check for normality by adding Skewness and Kurtosis to check for normality and asymmetry of the variables distributions. The enrichment of the interpretation of the results is well explained and the authors added an analysis of the heterogeneity results based on previous studies. The authors improve the results policy implications to section VIII.

The rectifications of the interpretations, discussions and policy implications contribute to the overall improvement of the manuscript. I recommend accepting the manuscript for publication.

7. PLOS authors have the option to publish the peer review history of their article (what does this mean?). If published, this will include your full peer review and any attached files.

Reviewer #1: No

Reviewer #2: No

---

## [Editor Report · Acceptance letter]

4 Dec 2023

PONE-D-23-15681R1 

Impact of digital finance on enterprise green innovation: From the perspective of information asymmetry, consumer demand and factor market distortions 

Dear Dr. Zhang:

I'm pleased to inform you that your manuscript has been deemed suitable for publication in PLOS ONE. Congratulations! Your manuscript is now with our production department. 

Kind regards, 

on behalf of

Dr. Zakaria Boulanouar 

Academic Editor

PLOS ONE